**TECHNIQUE**

# Unlocking the full potential of high-density surface EMG: novel non-invasive high-yield motor unit decomposition

Agnese Grison[1] , Irene Mendez Guerra[1], Alexander Kenneth Clarke[1], Silvia Muceli[1,2] ,
Jaime Ibáñez[1,3,4] and Dario Farina[1]

[1]*Department of Bioengineering, Imperial College, London, UK*
[2]*Department of Electrical Engineering, Chalmers University of Technology, Gothenburg, Sweden*
[3]*BSICoS, IIS Aragón, Universidad de Zaragoza, Zaragoza, Spain*
[4]*Centro de Investigación Biomédica en Red en Bioingeniería, Biomateriales y, Nanomedicina (CIBER-BBN), Zaragoza, Spain*

Handling Editors: Richard Carson & Madeleine Lowery

The peer review history is available in the Supporting Information section of this article (https://doi.org/10.1113/JP287913#support-information-section).

The Journal of Physiology

**Abstract figure legend** Schematic of Swarm-Contrastive Decomposition. A set of separation vectors are initialised to the same vector, with zero-mean and unitary standard deviation. Each separation vector is associated with a specific particle, the value of the exponent in the contrast function. This value tunes the sensitivity of the contrast function to outliers. Each separation vector is updated independently given a measure of sparsity of the produced source vector. The exponents are updated based on the optimal value with a particle swarm optimisation algorithm. The final outputs are the peaks of the source vector, which represent the firing times of the current motor unit. The process is repeated in an iterative fashion until all motor units are decomposed. Created in BioRender. Grison, A. (2025) https://BioRender.com/y04h318

This article was first published as a preprint. Grison A, Mendez Guerra I, Kenneth Clarke A, Muceli S, Ibanez Pereda J, Farina D. 2024. Unlocking the full potential of high-density surface emg: novel non-invasive high-yield motor unit decomposition. arXiv. https://doi.org/10.48550/arXiv.2410.14800

**Abstract** The decomposition of high-density surface electromyography (HD-sEMG) signals into motor unit discharge patterns has become a powerful tool for investigating the neural control of movement, providing insights into motor neuron recruitment and discharge behaviour. However, current algorithms, while effective under certain conditions, face significant challenges in complex scenarios, as their accuracy and motor unit yield are highly dependent on anatomical differences among individuals. To address this issue, we recently introduced Swarm-Contrastive Decomposition (SCD), which dynamically adjusts the contrast function based on the distribution of the data. Here, we demonstrate the ability of SCD in identifying low-amplitude motor unit action potentials and effectively handling complex decomposition scenarios. We validated SCD using simulated and experimental HD-sEMG recordings and compared it with current state-of-the-art decomposition methods under varying conditions, including different excitation levels, noise intensities, force profiles, sexes and muscle groups. The proposed method consistently outperformed existing techniques in both the quantity of decoded motor units and the precision of their firing time identification. Across different simulated excitation levels, SCD detected, on average, 25.9 $\pm$5.8 motor units *vs.* 13.9 $\pm$ 2.7 found by a state-of-the-art baseline approach. Across noise levels, SCD detected 19.8 $\pm$ 13.5 motor units, compared to 11.9 $\pm$ 6.9 by the baseline method. In simulated conditions of high synchronisation levels, SCD detected approximately three times as many motor units compared to previous methods (31.2 $\pm$ 4.3 for SCD, 10.5 $\pm$ 1.7 for baseline), while also significantly improving accuracy. These advancements represent a step forward in non-invasive EMG technology for studying motor unit activity in complex scenarios.

(Received 22 October 2024; accepted after revision 26 February 2025; first published online 17 March 2025)

**Corresponding author** D. Farina: Department of Bioengineering, Imperial College, London, UK. Email: d.farina@imperial.ac.uk

## Key points

- High-density surface electromyography (HD-sEMG) decomposition provides information on how the nervous system controls muscles, but current methods struggle in complex conditions.
- Swarm-Contrastive Decomposition (SCD) is a new approach that dynamically adjusts how signals are separated, improving accuracy and increasing the sample of detected motor units.
- SCD successfully identifies more motor units, including those with low-amplitude signals, and performs well even in challenging conditions such as high-interference signals.
- In simulated ballistic contractions, SCD detected three times more motor units than previous methods while improving accuracy.
- These advancements could improve non-invasive studies of muscle function in movement, fatigue and neurological disorders.

**Agnese Grison** obtained her MEng degree in Biomedical Engineering from Imperial College London. She is currently enrolled as a PhD candidate at the UKRI Centre for Doctoral Training in AI for Healthcare at Imperial College London. Her PhD research focuses on EMG decomposition and neural interfaces.

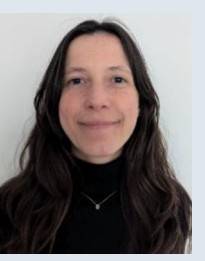

## Introduction

The nervous system regulates muscle force by transmitting signals from alpha motor neurons in the spinal cord to muscle fibres. The collective activity of these motor neurons, known as the neural drive to the muscle, determines the overall muscle activation and force production (de Luca & Erim, 1994; Farina & Negro, 2015). Each motor neuron's discharge triggers an action potential in each muscle fibre it innervates (Kandel et al., 2000). The sum of the action potentials of the fibres innervated by a single motor neuron is the motor unit action potential (MUAP). The sum of the MUAP trains from all active motor units generates the electromyography (EMG) signal, which represents the electrical activity of the muscle during a contraction (de Luca, 1979). Therefore, the EMG signal comprises both a neural component – the motor neuron discharge times – and a peripheral/muscle component – the MUAP waveforms (Stashuk, 2001). EMG decomposition aims to separate these two components, enabling the precise identification of the discharge times of each motor neuron.

For nearly a century, the primary method for investigating motor units has been through invasive EMG techniques, involving the decomposition of recordings from needle or wire electrodes under highly controlled conditions (Adrian & Bronk, 1929; Farina & Gandevia, 2024). These recordings have a high degree of selectivity, typically allowing the decoding of only a few motor units at a time. While high selectivity has been the primary condition to allow for the identification of individual motor unit activity from multi-unit recordings, this property also limits the number of detectable motor units. As a result, much of our current understanding of human motor neuron function is derived from a narrow set of motor units per subject. While these studies have been critical in establishing foundational principles, they do not provide a broad view across larger motor unit samples or more varied conditions. Surface EMG (sEMG) is less selective than invasive EMG and therefore traditionally lacked the precision required to isolate individual motor units.

The challenge of decomposing poorly selective sEMG recordings has been partially addressed by increasing the number of recording sites (electrodes) and applying blind source separation (BSS) techniques. Increasing electrode counts led to high-density sEMG (HD-sEMG) (Merletti & Muceli, 2019) which provided multiple observation points for the motor unit activities. Most BSS methods use the principles of independent component analysis (ICA) to separate sources (i.e. the motor units) from the observed mixtures. This is achieved through the use of a contrast function, a non-linear measure of sparsity of the signals, that is applied to the estimated spike trains in an iterative fashion (Farina & Holobar, 2016).

Because the HD-sEMG signals are convolutive mixtures of the motor neuron spike trains, decomposition of these signals requires convolutive BSS (cBSS). In cBSS, spike trains are extracted by optimising a contrast function that maximises an approximation of the skewness or kurtosis of the sources.

While BSS of HD-sEMG has greatly improved the study of motor units using non-invasive recordings, several challenges remain in its practical application. One major issue is that the number of successfully decomposed motor units can vary greatly across conditions, muscles and individuals (del Vecchio et al., 2020). This variability stems from the differences across conditions in the distribution of MUAPs at the skin surface, which is an inherent limitation of surface recordings. Indeed, in some scenarios, HD-sEMG decomposition may not be feasible at all due to these challenges. As a result, non-invasive motor unit investigations often focus on a narrow set of muscles, specific experimental subjects and controlled recording conditions (de Oliveira et al., 2022).

The underlying reason for poor HD-sEMG decomposition is that, in certain cases, the MUAPs from different motor units exhibit highly similar waveforms in both space and time, making it difficult for conventional contrast functions used in BSS to isolate them. This similarity limits the effectiveness of traditional decomposition algorithms. A promising direction for overcoming this limitation is the development of methods that enhance the differences between MUAPs by implementing novel strategies for blindly determining the optimal contrast function during decomposition. These approaches could improve the differentiation of similar MUAPs, leading to an increased number of decomposed motor units, improved classification accuracy and broader applicability of sEMG decomposition in more challenging scenarios.

In a recent study (Grison et al., 2024) we introduced Swarm-Contrastive Decomposition (SCD) for decomposing multi-channel intramuscular EMG recordings. The method optimises the contrast function dynamically for each source, increasing the separation between sources. Here, we use the concepts of SCD and we adapt it to the decoding of surface recordings.

By dynamically adapting the contrast function, we ensure that it is specifically tailored to the unique characteristics of the source being decomposed, significantly enhancing source separation performance. The adaptability of the contrast function distinguishes SCD from other state-of-the-art methods, which typically fix the contrast function to polynomial approximations of the kurtosis or the skewness (Holobar & Zazula, 2003; Negro et al., 2016). Moreover, SCD implements a peel-off strategy for sequential source removal, which allows the algorithm to detect smaller or more subtle sources that

would have otherwise been overshadowed by the larger, more dominant ones.

The proposed method was extensively validated with both simulated signals and experimental recordings across multiple conditions, including varying excitation levels, noise intensities, MUAP overlap, muscle groups and sexes. In all analyses, SCD was compared against current cBSS approaches, and specifically the method proposed by Negro et al. (2016), which is a current representative state-of-the-art decomposition approach. Henceforth, in the following we will refer to cBSS as to this particular method, representing one of the several solutions proposed to address the general convolutive demixing problem. The results demonstrated a substantial increase in performance of SCD in all conditions tested, proving the effectiveness of the approach in increasing both the number and the accuracy of the decomposed motor units. These findings indicated that, while the new method does not fully eliminate all challenges, SCD marks a significant advancement in HD-sEMG decomposition. By increasing the yield and accuracy of decomposition, it broadens the recording conditions and experimental scenarios where reliable motor unit activity can be extracted, ultimately increasing the practical utility and robustness of sEMG for a wide variety of applications.

## Methods

### EMG generation model

The EMG signal can be modelled using a linear time-invariant multiple-input–multiple-output system, which can be compactly expressed in matrix form as:

$$\boldsymbol{x}(t) = \sum_{l=0}^{L-1} \boldsymbol{H}(l)\,\boldsymbol{s}(t-l) + \boldsymbol{\xi}(t)$$

where $\boldsymbol{x}(t) = [x_1(t), x_2(t), \ldots, x_M(t)]^T$ is the vector of $M$ EMG channels, $\boldsymbol{s}(t) = [s_1(t), s_2(t), \ldots, s_N(t)]^T$ represent the $N$ motor unit spike trains and $\boldsymbol{\xi}(t)$ accounts for the additive noise. The matrix $\boldsymbol{H}(l)$ has size $M \times N$ and contains the $l^{\text{th}}$ sample of the $L$-sample-long MUAP waveforms that appear for each of the $N$ motor units across $M$ channels, assuming constant shape under stationary conditions. As this method assumes the repeatability of motor unit action potential shapes, it is not applicable in scenarios where the waveform shapes exhibit non-stationary variations, such as due to sliding of the electrodes, fatigue, movement of the joints or clinical conditions.

The convolutional model can be transformed into an instantaneous mixture by augmenting the vector of sources to include the $N$ original sources and their respective delayed versions. $L$ represents the duration of the impulse response of the filter, which models the volume conductor. To ensure a favourable ratio between the number of observations, that is the signals recorded at each electrode location, and sources, the observations are also extended by $R$ delayed versions. This reformulated instantaneous model is expressed as:

$$\tilde{\boldsymbol{x}}(t) = \tilde{\boldsymbol{H}}\tilde{\boldsymbol{s}}(t) + \tilde{\boldsymbol{\xi}}(t)$$

where $\tilde{\boldsymbol{s}}(t)$ is the $N(L+R) \times 1$ matrix constructed as:

$$\tilde{\boldsymbol{s}}(t) = \left[\tilde{\boldsymbol{s}}_1(t), \tilde{\boldsymbol{s}}_2(t), \ldots, \tilde{\boldsymbol{s}}_j(t), \ldots, \tilde{\boldsymbol{s}}_N(t)\right]^T$$

$$\tilde{\boldsymbol{s}}_j(t) = \left[s_j(t), s_j(t-1), \ldots, s_j(t-(L+R-1))\right]$$
$$j = 1, \ldots, N$$

and $\tilde{\boldsymbol{H}}$, of size $M(R+1) \times N(L+R)$, is constructed from the extended convolution kernels $\tilde{\boldsymbol{h}}$. The $M(R+1) \times 1$ observed signals $\tilde{\boldsymbol{x}}(t)$ are:

$$\tilde{\boldsymbol{x}}(t) = [\tilde{\boldsymbol{x}}_1(t), \tilde{\boldsymbol{x}}_2(t), \ldots, \tilde{\boldsymbol{x}}_i(t), \ldots, \tilde{\boldsymbol{x}}_M(t)]^T$$

$$\tilde{\boldsymbol{x}}_i(t) = [x_i(t), x_i(t-1), \ldots, x_i(t-R)] \quad i = 1, \ldots, M$$

In the absence of noise, the retrieval of discharge timings $\tilde{\boldsymbol{s}}(t)$ can be framed as the following inverse problem:

$$\tilde{\boldsymbol{s}}(t) = \boldsymbol{B}\tilde{\boldsymbol{x}}(t)$$

where $\boldsymbol{B}$ represents the approximate pseudo-inverse of $\tilde{\boldsymbol{H}}$. The goal of the decomposition process is to determine the $N$ separation vectors forming the matrix $\boldsymbol{B}$.

### Decomposition

We propose the use of the SCD decomposition algorithm (Grison et al., 2024) and compare it with a representative state of the art approach for cBSS decomposition (Negro et al., 2016).

Both algorithms rely on ICA to separate sources by maximising a statistical measure of non-Gaussianity or sparsity of the estimated sources (Hyvärinen & Oja, 2000). In BSS of EMG signals, the main characteristic used for decomposition is sparseness (Farina & Holobar, 2016) since motor unit discharge times are not fully independent. The non-linear function used to assess sparseness is called the contrast function, $G$. Similar to the Gaussianity property, mixing sources results in a signal less sparse than the individual sources. Thus, sparse sources can be identified by finding projections that maximise sparseness, making the choice of contrast function critical for the stability of the numerical optimisation process. Cumulants are a widely used class of non-linearities for this purpose. By optimising the separation vector to maximise higher-order cumulants, such as skewness or kurtosis, non-Gaussianity as well as

sparseness are amplified, increasing the likelihood of isolating sparse sources.

In the cBSS methods used for comparison (Negro et al., 2016) the contrast function is defined to maximise an approximation of the third cumulant (i.e. the skewness) of the sources, utilising $G_{cBSS}(s) = \frac{1}{6}s^3$. In contrast, our approach proposes to maximise an adaptively tuned higher-order cumulant for each source. Specifically, $G_{SCD}(s) = E\{sign(s)|s|^e\}$, where $e$ represents the exponent of the polynomial function, and $E$ is the expectation operator. The core hypothesis is that each source requires a different level of selectivity to be effectively separated from the mixture. Sources that are more similar to one another may require stronger discriminants, whereas others may require less stringent criteria. This adaptivity is the main difference between cBSS and the algorithm SCD.

In SCD, a candidate separation vector was randomly initialised from a zero-mean normal distribution with a standard deviation of 1. The candidate vector was then repeated for the number of initialised particles to produce the initial separation vectors $\boldsymbol{b}$. On each step, an ICA run was conducted on each separation vector independently for a maximum of 1000 iterations, with an early termination criterion applied if, after 20 iterations, $G_{SCD}(s)$ ceased to increase. After each ICA step, a peak-finding algorithm detected the source samples as calculated with the updated separation vectors, followed by a two-class $k$-medoid clustering applied to the estimated sources to identify their peaks as potential motor unit spikes. Source quality was evaluated using a fitness function based on the coefficient of variation of the interspike intervals, where the candidate with the lowest coefficient of variation was selected as the best source. The coefficient of variation was chosen as an appropriate metric due to the regularity of motor unit firing in isometric contractions. When regularity of the firings could not be assumed, as in the ballistic contractions, the silhouette measure (Negro et al., 2016) was used instead to assess source quality. After each ICA update, the optimal exponent for the contrast function was chosen from the current pool of candidates and updated by moving toward the exponent that yielded the highest-quality source. The separation vectors were then reinitialised with the spike-triggered average of the highest-quality source. Optimisation was halted after 10 updates of the optimal exponent coefficient. To assess the final source quality, the silhouette measure was used to evaluate the clustering separation between the decomposed source and the background noise.

If the source passed this evaluation (silhouette > 0.85) and had not been identified in previous iterations, the motor unit action potentials were subtracted (i.e. peeled off) from the signal by creating a template motor unit waveform and removing it at each identified motor unit firing time. This subtraction prevented further convergence to the same source in subsequent decomposition iterations.

However, from the estimated discharge times, motor units exhibiting discharge properties outside the pre-defined thresholds – specifically, a coefficient of variation above 35% or a firing rate outside the range 2–35 Hz (Martinez-Valdes et al., 2017) – were excluded from further analysis.

To assess the importance of the two core features of SCD – the adaptive contrast function and the peel-off procedure – two targeted ablations were conducted. The first ablation focused on addressing repeated convergence to the same source. Three strategies were tested: (1) initialising the separation vectors $\boldsymbol{b}$ with the activity index, a proxy for global pulse train activity (Holobar & Zazula, 2007) and eliminating the peel-off process, while preventing previously accepted motor unit firing timings from being reinitialised by excluding them from the activity index; (2) combining the activity index with a source deflation method to enforce orthogonality between separation vectors; and (3) comparing with the proposed SCD.

The second ablation maintained the peel-off method across all configurations and examined the effect of different exponents in the contrast function. These were compared against the performance of SCD with the dynamically adapted exponents.

The decompositions were run on an Intel(R) Core™ i7-10700K CPU with an Nvidia RTX 3080 GPU.

## Simulations

HD-sEMG data were simulated using NeuroMotion (Ma, Mendez Guerra et al., 2024) an advanced EMG simulator designed to produce physiological electric potentials during voluntary forearm movements. NeuroMotion operates through three modules. The first, an upper-limb musculoskeletal model developed using the OpenSim API (Delp et al., 2007), defines and visualises movements while estimating muscle fibre lengths and muscle activation levels. These estimates are then input into BioMime (Ma, Clarke et al., 2024) an AI-based volume conductor model that generates MUAPs based on parameters such as fibre number, depth, angular position, innervation zone and conduction velocity, derived from a myoelectric digital-twin model (Maksymenko et al., 2023). The final module, a motor unit pool model, converts neural inputs into spike train simulations, completing the muscle activation simulation (Fuglevand et al., 1993).

An isometric and isotonic index finger contraction from the flexor digitorum superficialis was simulated with NeuroMotion. The simulated signals were generated for an electrode band comprising a $10 \times 32$ electrode grid, positioned around the proximal third of the forearm.

Figure 1*A* displays a schematic of the simulated setup. A pool of 100 motor neurons was simulated. The innervated muscle fibres were randomly and uniformly distributed throughout the muscle volume, with innervation zones normally distributed around 50 ± 10% of the total fibre length. An exponential function was employed to model the recruitment thresholds and number of innervated fibres for each motor neuron, resulting in a higher proportion of small, low-threshold motor neurons compared to larger, high-threshold ones.

Conduction velocities were sampled from a normal distribution (mean 4.0 ± 0.5 ms$^{-1}$, truncated between 3 and 4.5 ms$^{-1}$ to avoid non-physiological parameters) and sorted based on the number of innervated fibres. Motor neurons began firing at a baseline rate of 8 pulses per seconds (pps) once the excitation level surpassed their recruitment threshold (Fuglevand et al., 1993). The discharge rate increased linearly by 3 pps for every 10% increase in excitation (Fuglevand et al., 1993; Keenan et al., 2006). Maximum discharge rates varied from 35 pps for the first recruited motor neuron to 25 pps for higher-threshold motor neurons (Fuglevand et al., 1993). The final motor neuron was recruited at 50% of the maximum excitation level (Fuglevand et al., 1993). The variability in discharge times followed a Gaussian random process with a coefficient of variation of 0.2 (Fuglevand et al., 1993; Keenan et al., 2006).

While the myoelectric digital-twin model (Maksymenko et al., 2023) provided the muscle geometry based on magnetic resonance imaging, NeuroMotion estimated other motor unit parameters (position, length, depth, angle, innervation zone and conduction velocity) based on physiological ranges. These distributions were randomised over 10 bootstrapping iterations to capture the full variability observed in real data ($N = 10$).

Three distinct analyses were performed on different types of simulated data to investigate the characteristics of the proposed algorithm. The first analysis examined the effect of excitation level on the decomposition process by gradually increasing the excitation from 10 to 100% of maximal voluntary contraction (MVC) in 10% increments, forming a dataset of 100 recordings, given by 10 bootstrapping iterations for each of the 10 force levels. For each MVC level, each contraction lasted 30 s. Throughout this analysis, the signal to noise ratio was maintained at 25 dB (zero-mean white Gaussian noise

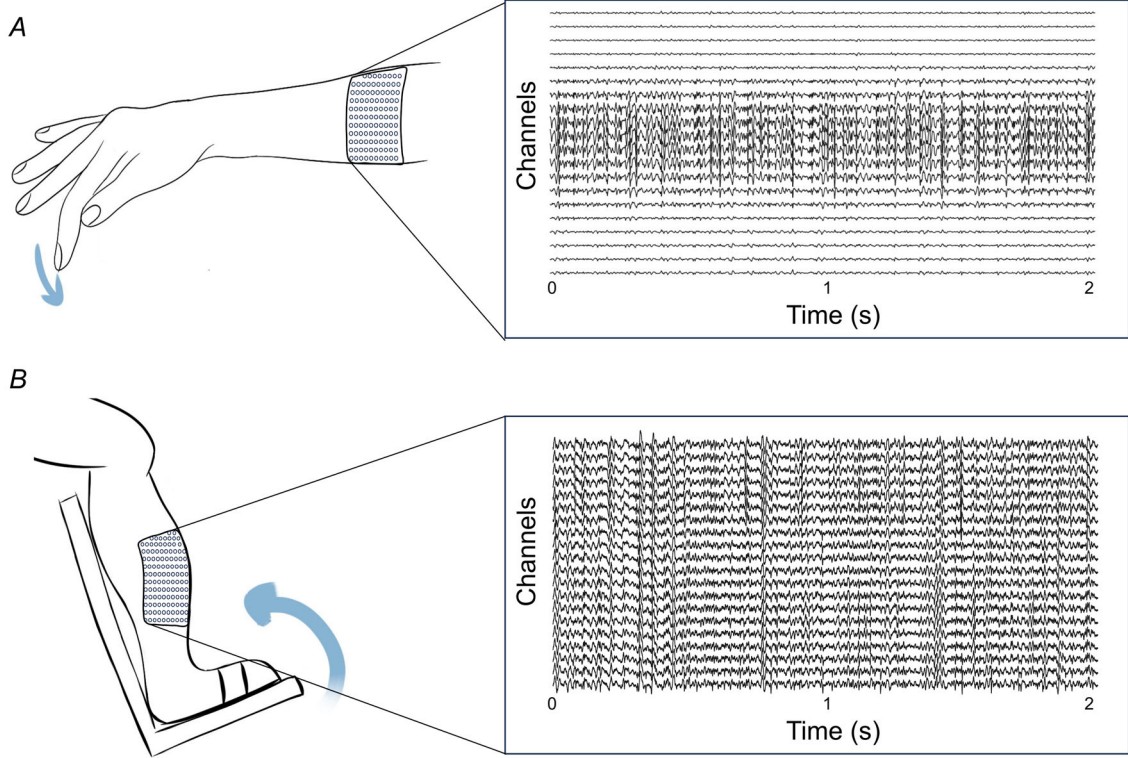

**Figure 1. Schematics of the HD-sEMG data used in the analysis**
*A*, simulated HD-sEMG data from forearm muscles during index finger flexion, with the electrode grid positioned over the proximal third of the forearm. The same setup was applied for the experimental recordings from the forearm. Representative data (simulations) are shown for a 30 %MVC contraction. *B*, experimental HD-sEMG data were also recorded from the TA muscle during ankle dorsiflexion. Representative data for a 20 %MVC contraction are displayed. HD-sEMG, high-density surface electromyography; MVC, maximal voluntary contraction; TA, tibialis anterior. [Colour figure can be viewed at wileyonlinelibrary.com]

with standard deviation based on the amplitude of the originally simulated noiseless EMG signals across time and channels) to isolate the effect of the excitation level.

The second analysis investigated the impact of the noise level on motor unit decomposition. In these simulations, the excitation level was fixed at 30 %MVC during 30 s isometric and isotonic contractions, while the noise level was varied from 10 to 30 dB, with 5 dB increments for 10 bootstrapping iterations ($N = 10$). This approach allowed for examination of the performance of the algorithm under different levels of signal contamination, simulating more challenging real-world conditions.

The last analysis focused on ballistic contractions (high synchronisation level), where the force increased rapidly from 0 to 40 %MVC. Thirty bursts of isometric muscle activity were simulated for 10 bootstrapping iterations ($N = 10$), each lasting 1 s and separated by 3 s intervals. These conditions mimicked sudden, explosive muscle contractions, enabling evaluation of the performance of the algorithm under high levels of synchronised motor unit activity. The noise level in this experiment was maintained at 25 dB. It is noteworthy that while the contraction was dynamic in terms of force, the posture and muscle geometry remained constant throughout the simulation.

### Experimental data

All reported experiments adhered to the ethical guidelines set by Imperial College London (ICREC Project ID 19IC5640). All procedures were conducted in accordance with the *Declaration of Helsinki*, with informed consent obtained from the participants prior to each experiment.

Three experiments were conducted, with the first two focusing on the tibialis anterior (TA) muscle of healthy male and female participants respectively, and the last one focusing on the forearm of healthy female participants. In the TA experiments, participants were seated with their right leg and foot secured to a dynamometer and they were instructed to sustain an isometric and isotonic ankle dorsiflexion. Figure 1*B* displays a schematic of the experimental setup. Similar to the simulated conditions, subjects in the forearm experiments were instructed to perform isometric index finger flexion (Fig. 1*A*). The force levels were set as percentages of the participants' MVC, with visual feedback provided for both exerted force and target. EMG signals were recorded in monopolar derivation, with a reference electrode placed on the ankle (TA), and wrist (forearm), and bandpass-filtered between 20 and 500 Hz for the signals collected at the surface, and between 20 and 4400 Hz for the intramuscular signals. All signals and force data were recorded concurrently using the Quattrocento multi-channel amplifier (OT-Bioelettronica, Torino, Italy), featuring a high common mode rejection ratio >95 dB, high-pass filtered at 10 Hz, and digitised at 16-bit resolution.

**Pilot 1: two-source validation (TA).** Two healthy men, aged 39 and 30 years, were recruited ($N = 2$). The protocol included 20 s isometric and isotonic contractions at 10, 20 and 30 %MVC, a 15 s contraction at 40 %MVC, and 10 s contractions at 50, 60 and 70 %MVC. Three 40-channel HD intramuscular EMG (HD-iEMG) micro-electrode arrays (Muceli et al., 2015, 2022) were implanted in the TA, oriented longitudinally and spaced ~3 cm apart. The electrodes, made of platinum, are arranged on two sides of a wider filament, with each side consisting of 20 electrodes spaced 1 mm apart. The two sides are offset by 0.5 mm (Muceli et al., 2022). Two 64-channel HD-sEMG grids (4 mm inter-electrode distance, 13 × 5 electrode configuration) were placed on the skin surface above the intramuscular detection sites. Intramuscular and surface EMG signals were concurrently sampled at 10,240 Hz. The concurrent recording of intramuscular and surface EMG signals allowed for the use of the two-source validation methods for an objective and rigorous assessment of decomposition accuracy (Farina, Merletti et al., 2014; Farina, Negro et al., 2014; Mambrito & de Luca, 1984).

**Pilot 2: sex differences in motor unit yield (TA).** Two healthy women, aged 23 and 24 years, were recruited ($N = 2$). The protocol included 20 s isometric and isotonic contractions at 5 and 10 %MVC. One 256-channel HD-sEMG grid (4 mm inter-electrode distance, 32 × 8 electrode configuration) was placed on the TA. HD-sEMG signals were sampled at 2048 Hz. The measurements on female individuals allowed us to compare the algorithms in conditions that are challenging for sEMG decomposition since it has been reported that decomposition yield and accuracy decrease in female individuals (del Vecchio et al., 2020; Lulic-Kuryllo & Inglis, 2022; Taylor et al., 2022).

**Pilot 3: motor unit yield in complex muscle groups (forearm).** Two healthy women, aged 26 and 30 years, were recruited for this study ($N = 2$). The protocol included 20 s isometric and isotonic index finger flexions at 15 %MVC. Three 64-channel HD-sEMG grids (8 mm inter-electrode distance, 13 × 5 electrode configuration) were placed around the proximal third of the forearm of each participant. HD-sEMG signals were sampled at 2048 Hz. The measurements on the forearm allowed us to analyse the performance of the algorithms in more complex muscle groups, where the number of motor units decomposed is usually limited to fewer than 10 (del Vecchio et al., 2020).

## Metrics for accuracy

The level of agreement between the discharge times of motor units as decoded from the decomposition of the simulated HD-sEMG and the ground truth was assessed using the rate of agreement (RoA). The RoA measures the fraction of commonly identified discharges relative to the total number of discharges, considering both common and not-common discharge times. The RoA was therefore calculated as follows:

$$RoA = \frac{TP}{TP + FP_1 + FP_2}$$

where $TP$ refers to the number of matched predicted activations within a deviation margin of $\pm 0.5$ ms (Farina et al., 2001). $FP_1$ and $FP_2$ represent the counts of unmatched predicted activations, corresponding to discharge times present in only one of the two sets.

Several methods have been proposed to assess the accuracy of decomposition of experimental HD-sEMG data (Holobar et al., 2010, 2014; Hu et al., 2013) with the two-source validation being the most reliable (Farina, Merletti et al., 2014, Farina, Negro et al., 2014; Mambrito & de Luca, 1984). This approach involves recording both HD-sEMG and HD-iEMG signals concurrently, decomposing them independently, and comparing the discharge times from the two decompositions to determine the RoA, as defined above. This approach operates on the principle that similar results from two independent algorithms applied to different signals are probably correct, as the probability of identical errors is low. The RoA thus reflects the relative performance of the algorithms without bias toward either method (Farina, Merletti et al., 2014, Farina, Negro et al., 2014). Here, when available, we utilised the concurrent recording of the HD-iEMG micro-electrode arrays to validate the surface decomposition. The HD-iEMG data were decomposed using SCD, as described in Grison et al. (2024). When the concurrent recording of the HD-iEMG signals was not available, the number of motor units decoded by SCD and cBSS was compared, and the RoA between the commonly identified motor units (by SCD and cBSS) was also analysed.

For all analyses, the RoA is reported in percentages as the mean and standard deviation. Additionally, the number of motor units is reported as the mean and standard deviation across bootstrapping iterations.

## Statistical analysis

All statistical analyses were conducted in Python 3.10 using the SciPy library (Virtanen et al., 2020) and the Pingouin statistical package. A significance threshold of $P < 0.05$ was applied to all tests.

Two-way repeated-measures ANOVAs were performed to assess the effects of independent factors (MVC levels, noise levels, the exponent used in the contrast function, and the method for preventing repeated source convergence) on the dependent variable (i.e. the number of decomposed motor units), including their interaction effects. *Post hoc* pairwise comparisons with Bonferroni corrections were conducted to identify specific differences between factors.

To compare motor unit characteristics (MUAP peak-to-peak amplitudes, conduction velocities, number of fibres, depth and exponents of the contrast function), independent *t* tests were performed when normality was assumed (as determined by the Shapiro–Wilk test, $P < 0.05$); otherwise, Mann–Whitney *U* tests were used.

For comparing the RoA of motor units decomposed from HD-iEMG recordings and commonly identified by both SCD and cBSS, Wilcoxon signed-rank tests were applied due to non-normal distributions of the RoAs.

## Results

### Simulations: excitation level

Figure 2 summarises the results of motor unit decomposition using SCD and cBSS across varying excitation levels in the simulated data. Figure 2*A* illustrates the mean number of decomposed motor units over 10 bootstrapping iterations. Across all excitation levels and bootstrapping iterations, on average, cBSS identified $13.9 \pm 2.7$ motor units per contraction, while SCD identified on average nearly a double number of motor units ($25.9 \pm 5.8$). Two-way repeated-measures ANOVA revealed significant main effects of the decomposition algorithm ($P < 0.001$) and MVC level ($P < 0.001$), as well as a significant interaction between the algorithms and MVC levels ($P < 0.001$). *Post hoc* analyses showed that SCD resulted in a significantly higher number of decomposed motor units than cBSS ($P < 0.001$). Of the motor units identified by cBSS, 98% were also detected by SCD, demonstrating that SCD primarily expanded the total count of detected units, rather than identifying significantly different units from those already captured by cBSS. Figure 2*B* reports the final exponent distribution of the automatically selected contrast functions for all identified sources with SCD. The RoA of all the motor units decomposed by SCD was $98.90 \pm 2.33\%$, compared to $98.02 \pm 2.69\%$ for cBSS. Figure 2*B* shows the RoA for the motor units identified by both methods. In these units, SCD showed a significant increase in RoA ($P < 0.001$). The effect size, as measured by a rank-biserial correlation of 0.99, indicates a substantial and practically meaningful improvement in RoA.

A comparison between the motor units which were detected by both SCD and cBSS ('common') and those

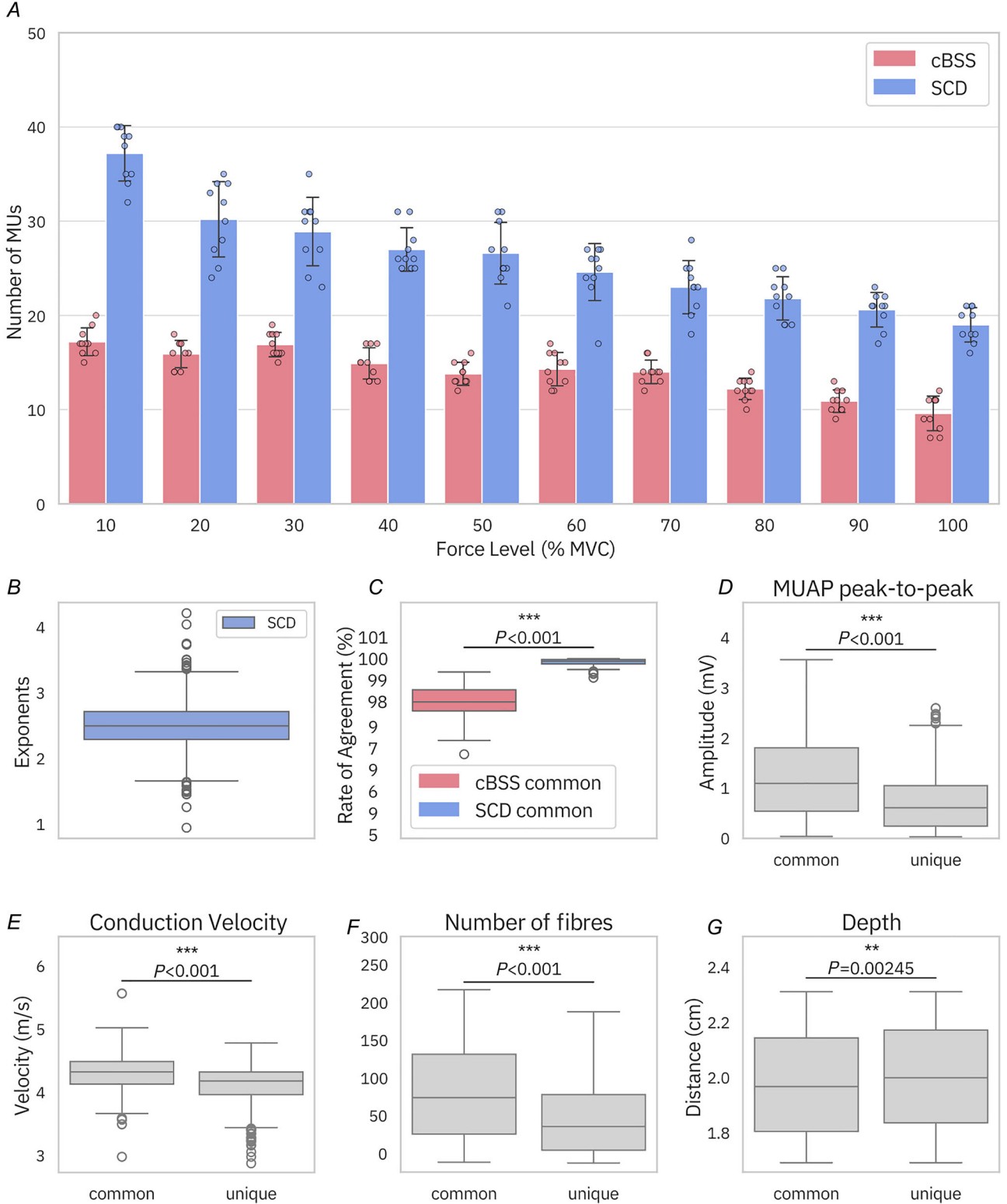

**Figure 2. Effect of excitation level on the number of motor units found and characteristics of the decomposed motor units**

*A*, mean number of motor units per bootstrap iteration for SCD (blue) and cBSS (pink). Error bars indicate the standard deviation across bootstrapping iterations. Individual data points from each bootstrapping iteration are overlaid on the bar chart. SCD decomposed significantly more motor units than cBSS ($P < 0.001$). *B*, distribution of exponents for SCD. *C*, distribution of the RoA between the automatic methods and the simulated ground

truth for the motor units commonly identified by SCD and cBSS. *D–G*, distributions of the peak-to-peak MUAP amplitudes (*D*), conduction velocity (*E*), number of fibres innervated (*F*) and depth (measured with respect to the skin, higher is deeper) (*G*) for motor units common to both SCD and cBSS, and those uniquely identified by SCD. cBSS, convolutive BSS; RoA, rate of agreement; SCD, Swarm-Contrastive Decomposition.
[Colour figure can be viewed at wileyonlinelibrary.com]

identified only by SCD ('unique') is presented across various motor unit properties. These properties include peak-to-peak amplitudes of the MUAP in the channel with the highest amplitude (Fig. 2*D*), motor unit conduction velocities (Fig. 2*E*), the number of fibres innervated per motor unit (Fig. 2*F*) and the depth of the motor units (Fig. 2*G*). These results show that the motor units uniquely identified by SCD tended to exhibit lower peak-to-peak amplitudes, suggesting that these motor units were either located farther from the electrodes or deeper within the muscle. However, Fig. 2*G* shows that, while significant, the difference in distance from the electrodes between common and unique units was relatively small. Moreover, the SCD-unique motor units typically had slower conduction velocities and a smaller number of innervated fibres, indicating they were smaller motor units.

The computational time required to run one decomposition was ~2 min for cBSS and 12 min for SCD.

## Ablations

Figure 3*A* displays the mean number of decomposed motor units across bootstrapping iterations, stratified by force level, for the three methods employed to mitigate repeated convergence to the same source during the ICA optimisation. On average, across force levels and bootstrapping iterations, the activity index method identified $8.4 \pm 2.6$ motor units, the deflation method $11.3 \pm 1.7$ motor units and the peel-off method $25.9 \pm 5.8$ motor units. Two-way repeated-measures ANOVA revealed significant main effects of the method used to prevent repeated convergence to the same source on the number of decomposed motor units. The analysis revealed significant main effects for both MVC ($P < 0.001$) and the method used (*$P < 0.001$), as well as a significant interaction between the two independent factors ($P < 0.001$). *Post hoc* pairwise comparisons revealed significant differences between the three methods in their ability to decompose motor units. The peel-off method consistently outperformed both the activity and deflation methods with highly significant differences ($P < 0.001$ for peel-off *vs.* activity; $P < 0.001$ for peel-off *vs.* deflation). These results demonstrate that the peel-off method identifies significantly more motor units than the other methods. Additionally, the deflation method also significantly outperformed the activity method ($P < 0.001$). These results indicate that both force level and method to

prevent convergence on the same source significantly influence the number of decomposed motor units.

To further understand the interaction between the ablation method and the way in which contrast functions are selected, we compared the number of units decomposed using the best ablation method in the previous test (peel-off) and four ways of defining the contrast functions for the separation: fixed contrast functions with exponents of orders 2, 2.4 and 3, and using SCD to optimise the selection of the contrast function used for each decomposed source (Fig. 3*B*). The fixed values were selected to look at the median value of the adapted swarm exponents ($e = 2.4$) and the closest integer lower ($e = 2$) and upper ($e = 3$) bounds. On average, across all force levels and bootstrapping iterations, the method with $e = 2$ identified $10.9 \pm 4.6$ motor units, the method with $e = 2.4$ identified $21.1 \pm 3.5$ motor units, the method with $e = 3$ identified $6.4 \pm 3.4$ motor units and the method with the adaptive exponent identified $25.9 \pm 5.8$ motor units. Two-way repeated-measures ANOVA revealed significant main effects of both MVC level ($P < 0.001$) and exponents used in the contrast functions ($P < 0.001$) on the number of decomposed motor units, as well as a significant interaction between the two factors ($P < 0.001$). *Post hoc* pairwise comparisons revealed significant differences between all four methods. The swarm method resulted in a significantly higher number of decomposed motor units compared to $e = 2$ ($P < 0.001$), $e = 2.4$ ($P < 0.001$) and $e = 3$ ($P < 0.001$).

These results reflect an important improvement in the decomposition when an adaptive method to select contrast functions is combined with the peel-off method for iterative source removal. Using either of these two configurations alone with either fixed contrast functions or other possible source removal methods resulted in drastic reductions in the number of decomposed motor units.

## Simulations: noise level

Figure 4 presents the mean number of decomposed motor units across bootstrap iterations at a fixed excitation level of 30 %MVC (Fig. 4*A*) and the RoA of the identified motor units compared to the simulated ground truth (Fig. 4*B*). Two-way repeated-measures ANOVA revealed significant main effects of both noise level ($P < 0.001$) and decomposition algorithm ($P < 0.001$) on the number of decomposed motor units, as well as a significant interaction between the two factors

($P < 0.001$). *Post hoc* pairwise comparisons revealed significant differences between all noise levels, with higher noise levels resulting in fewer decomposed motor units. Additionally, the SCD algorithm identified significantly more motor units than the cBSS algorithm across all noise levels ($P < 0.001$). These results indicate that both noise level and decomposition algorithm significantly influence the number of decomposed motor units, with SCD demonstrating superior performance. On average across all noise levels and bootstrapping iterations, cBSS identified $11.9 \pm 6.9$ motor units, while SCD detected $19.8 \pm 13.5$ motor units. Approximately 97% ($11.5 \pm 7.2$) of the motor units detected by cBSS were also detected by SCD. The RoA for the motor units identified by SCD was $99.1 \pm 1.8$%, compared to $98.2\% \pm 2.6\%$ for cBSS. The RoA of the motor units commonly identified by both SCD and cBSS (Fig. 4*B*) was significantly higher for SCD than for cBSS ($P < 0.001$).

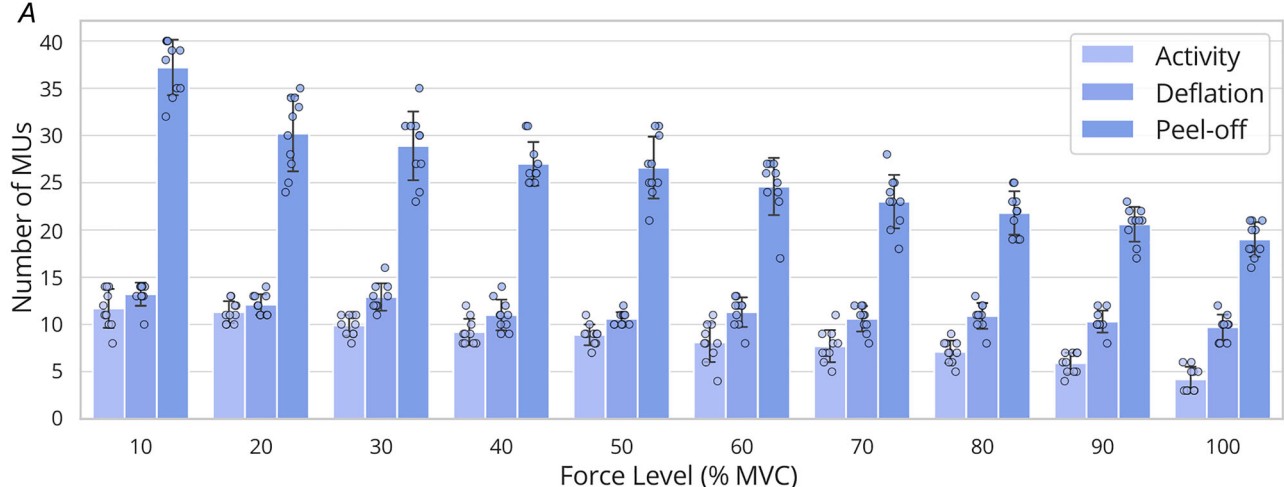

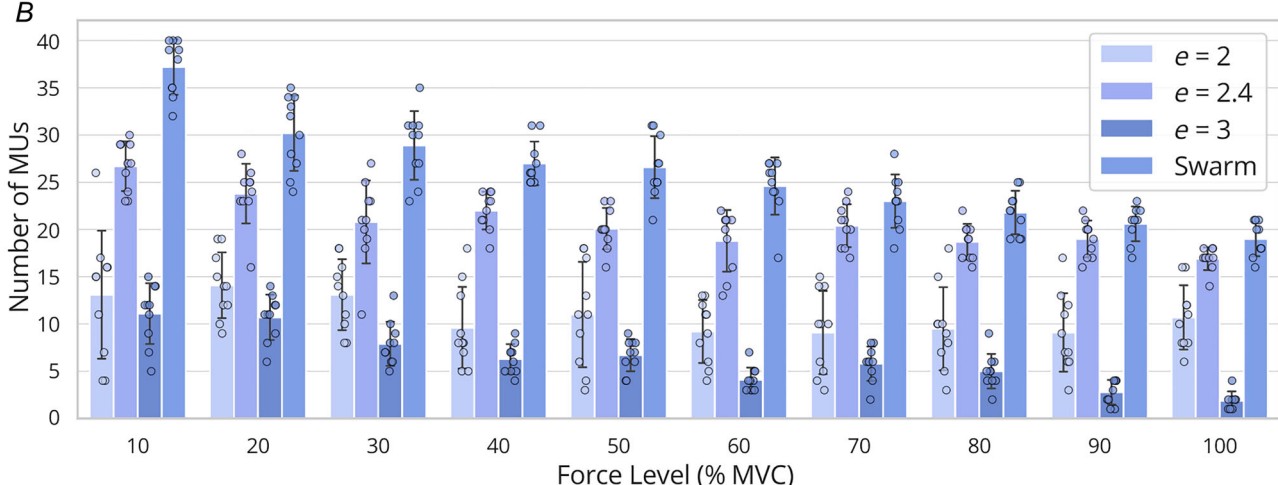

**Figure 3. Effect of method to prevent repeated convergence to the same source and exponent of the contrast function on the number of motor units found**

*A*, effect of the method used to prevent source convergence on the number of motor units found. Bars represent the mean of the bootstrapped samples, while error bars reflect the standard deviation across these bootstrapping iterations. Individual data points from each bootstrapping iteration are overlaid on the bar chart. Three methods are reported: (1) activity (use the activity index to initialise the separation vectors), (2) deflation (activity index to initialise the separation vectors and orthogonalise the separation vectors) and (3) peel-off (remove found sources from the EMG). The peel-off method decomposed significantly more motor units than the other two methods ($P < 0.001$). *B*, effect of the exponent of the contrast function on the number of motor units found. The peel-off approach was used for all the three methods reported: (1) exponent fixed at 2, (2) exponent fixed at 2.4, (3) exponent fixed at 3, and (4) exponents starting at [2, 3, 4, 5, 6, 7] and updated with particle swarm optimisation. The decomposition with the swarm update found significantly more motor units than fixing the exponent values ($P < 0.001$). EMG, electromyography. [Colour figure can be viewed at wileyonlinelibrary.com]

## Simulations: ballistic contractions

The decomposition performance was preserved in contractions simulated with ballistic force changes, where the overlap of the MUAPs over time was greater due to the rapid recruitment and de-recruitment of the motor units.

Figure 5 provides a representative example of the signal and the decomposed activity. Figure 5*A* presents the spike-triggered average of the MUAP of a decomposed unit. The estimated activity of the unit in one of the bursts of EMG is shown in Fig. 5*B*, which illustrates a clear separation between the source components (marked with red circles) and the background activity. A representative example of 10 channels of the simulated EMG for the same 1 s interval is depicted in Fig. 5*C*, reported for the specific spatial grid configuration (10 × 32).

Figure 6 presents the results of this analysis. Figure 6*A* shows the number of decomposed motor units for both cBSS and SCD, with SCD identifying approximately three times the number of motor units of cBSS. On average across the bootstrapping iterations, cBSS detected 10.5 ± 1.7 motor units, while SCD decoded 31.2 ± 4.3, which were statistically different ($P < 0.001$). On average, 96% of the motor units identified with cBSS were also identified with SCD. The RoA of all the motor units decomposed by SCD was 97.5 ± 3.6%, while cBSS achieved 97.1 ± 2.4%. Of the commonly identified motor units, SCD achieved a significantly higher RoA ($P < 0.001$). The distribution of RoA for the commonly

detected motor units is illustrated in Fig. 6*B*. Additionally, the motor units uniquely identified by SCD displayed significantly lower ($P < 0.001$) peak-to-peak MUAP amplitudes compared to those found by cBSS (Fig. 6*C*). This further demonstrates SCD's capability to decompose lower-amplitude motor units that are missed by cBSS.

## Pilot 1: two-source validation

This experiment aimed to compare the HD-sEMG decomposition results between SCD and cBSS using experimental data. The outputs from both methods were validated by comparing them against the decomposition of concurrently recorded HD-iEMG signals, providing a reliable benchmark for assessing the accuracy of the surface decompositions.

Across the force levels, SCD identified 41.6 ± 12.1 motor units for subject 1 and 12.0 ± 5.3 for subject 2, a significant increase ($P < 0.001$) with respect to cBSS (13.7 ± 3.1 and 2.1 ± 0.7 respectively). Additionally, the number of matched motor units between HD-iEMG and HD-sEMG was greater for SCD than for cBSS (Fig. 7*A*). This was probably due to the fact that SCD could identify deeper or smaller sources that were not separable with cBSS but that were captured by the intramuscular multi-electrode arrays.

To assess decomposition accuracy of the two compared methods, the RoA for the motor units found by both cBSS

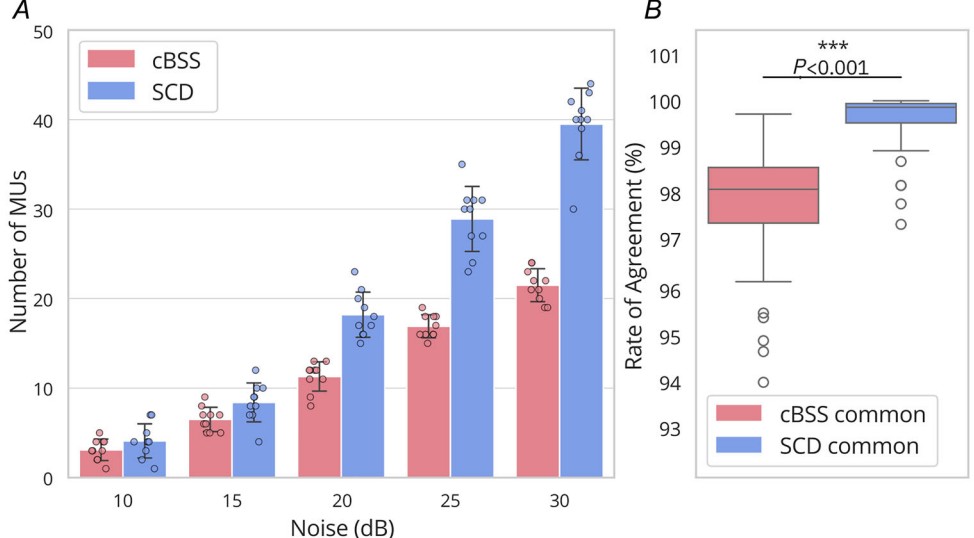

**Figure 4. Effect of noise on the number of motor units found and the RoA with the ground truth for 30 %MVC force level**

*A*, number of motor units against noise level, for SCD and cBSS. Bars represent the mean of the bootstrapped samples, while error bars reflect the standard deviation across these bootstrapping iterations. Individual data points from each bootstrapping iteration are overlaid on the bar chart. SCD decomposed significantly more motor units than cBSS ($P < 0.001$). *B*, distribution of the RoA between decomposed motor units and their simulated ground truth for the motor units commonly identified by SCD and cBSS. cBSS, convolutive BSS; MVC, maximal voluntary contraction; RoA, rate of agreement; SCD, Swarm-Contrastive Decomposition. [Colour figure can be viewed at wileyonlinelibrary.com]

and SCD, and matched with HD-iEMG, was calculated. The RoA for SCD was 97.3 ± 3.6%, compared to 95.0 ± 4.4% for cBSS.

Figure 7*B* shows the distribution of the RoA for the commonly identified motor units, which shows a shift toward higher values for SCD (although there was only a trend for this shift; $P = 0.08$). Additionally, when comparing the motor units identified by both cBSS and SCD, as well as those uniquely identified by SCD, it was observed that SCD successfully detected motor units with significantly lower peak-to-peak MUAP amplitudes ($P < 0.001$), underscoring its ability to identify smaller motor unit activity that may have been overlooked by cBSS.

Additionally, the exponent for HD-sEMG was significantly higher ($P < 0.001$) than that for HD-iEMG (Fig. 7*D*).

**Pilot 2: sex differences in motor unit yield (TA).** In this experiment, the performance of SCD and cBSS were compared when decomposing signals recorded from female subjects. Figure 8 shows the number of decomposed motor units per subject stratified by force

level (Fig. 8*A*) and the RoA between the motor units identified by the two methods (Fig. 8*B*). On average across subjects and force levels, SCD identified 16.8 ± 3.8 motor units (20 and 14 at 5 %MVC, and 20 and 13 at 10 %MVC for each subject respectively), while cBSS identified 10.0 ± 1.8 motor units (11 and 8 at 5 %MVC, and 12 and 8 at 10 %MVC for each subject respectively). Importantly, SCD identified all the motor units identified by cBSS, and additional ones. The RoA between SCD and cBSS was 97.2 ± 2.4% for 5 %MVC, and 95.9% ± 3.6% for 10 %MVC (Fig. 8*B*). The differences in the peak-to-peak amplitudes of the MUAPs between the units commonly found by both cBSS and SCD and those uniquely identified by SCD were not significant (Fig. 8*C*).

**Pilot 3: motor unit yield in complex muscle groups (forearm).** In this experiment, the performance of SCD and cBSS were compared when decomposing signals recorded from the forearm muscles of female subjects. Figure 9 shows the number of decomposed motor units (Fig. 9*A*) and the RoA between the motor units identified by the two methods (Fig. 9*B*). For each subject, SCD identified 17 and 22 motor units, while cBSS identified 10

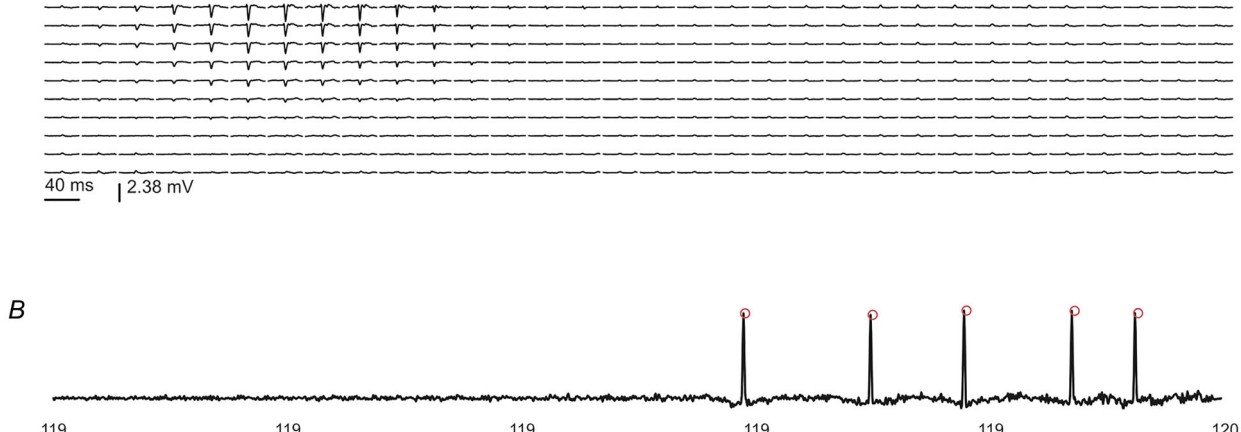

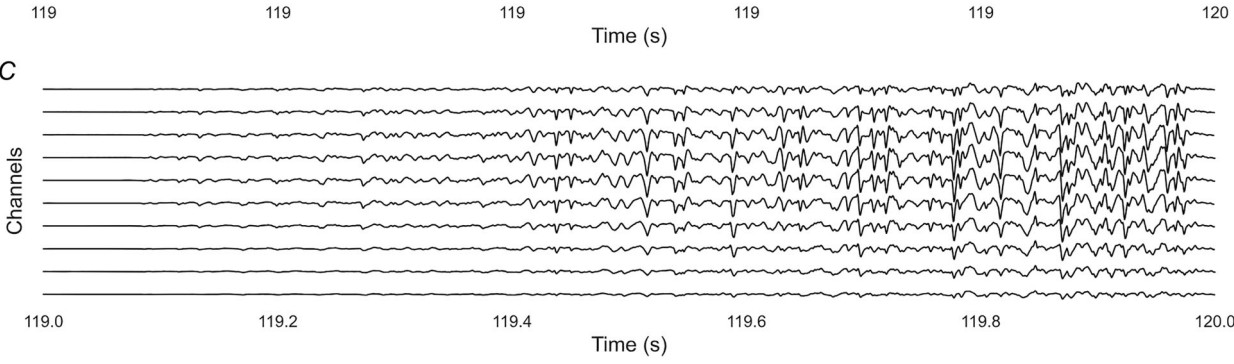

**Figure 5. Representative example of a unit in a ballistic contraction decomposed with 100% accuracy**
*A*, spatial distribution of the MUAP arranged in the 10 × 32 electrode configuration. *B*, a 1 s zoom-in of the innervation pulse train of the full source. The discharge times of the clustered source are shown in red. *C*, a 1 s zoom-in of the EMG. EMG, electromyography; MUAP, motor unit action potential. [Colour figure can be viewed at wileyonlinelibrary.com]

and 17 motor units respectively. Only four units that were identified by cBSS were not identified by SCD. Across the units with high level of agreement, the RoA between SCD and cBSS was 96.7 ± 2.6% (Fig. 9*B*). The differences in the peak-to-peak amplitudes of the MUAPs between the units commonly found by both cBSS and SCD and those uniquely identified by SCD were not significant (Fig. 9*C*).

## Discussion

We proposed and validated SCD for the decomposition of HD-sEMG signals, demonstrating its superior performance over state-of-the-art approaches. As demonstrated with both simulations and experimental results, SCD represents a major step forward in HD-sEMG decomposition, with broad implications for the study of the neural control of movement. The core strength of SCD lies in its dynamic adaptation of the contrast function via particle swarm optimisation and the incorporation of a peel-off strategy for sequential source removal. These features enable SCD to address critical challenges in HD-sEMG decomposition, such as differentiating motor units with highly similar MUAPs and identifying motor units with low-energy MUAPs. Unlike conventional BSS algorithms that rely on fixed contrast functions (Holobar & Zazula, 2003; Negro et al., 2016), the ability of SCD to

adjust its contrast function allows for a greater flexibility and improved handling of diverse signal characteristics.

SCD was validated using both simulated and experimental data. In simulations, SCD decomposed a number of motor units nearly double those identified by classic cBSS. The motor units identified by SCD had smaller MUAPs (Figs 2*D* and 7*C*), and were therefore either deeper in the muscle tissue or smaller (Fig. 2*G*). Accordingly, the motor units identified by SCD had slower conduction velocities and fewer innervated fibres (Fig. 2*E* and *F*).

When further applied to simulated data during ballistic tasks (Fig. 5*C*), SCD demonstrated a 3-fold increase in the number of decomposed motor units (Fig. 6*A*), while reaching higher accuracy than the state-of-the-art. Furthermore, simulations also demonstrated that SCD identified a greater number of motor units than cBSS at varying levels of noise, despite the difference between methods decreasing with an increase in noise (Fig. 4*A*).

The simulations also revealed that both the adaptive contrast function and the peel-off procedure significantly contributed to SCD's performance (Fig. 3). Interestingly, we showed that fixing the exponent to the median value of the adaptively determined exponents during the swarm update yielded significantly fewer decomposed motor units compared to allowing the exponent to adapt

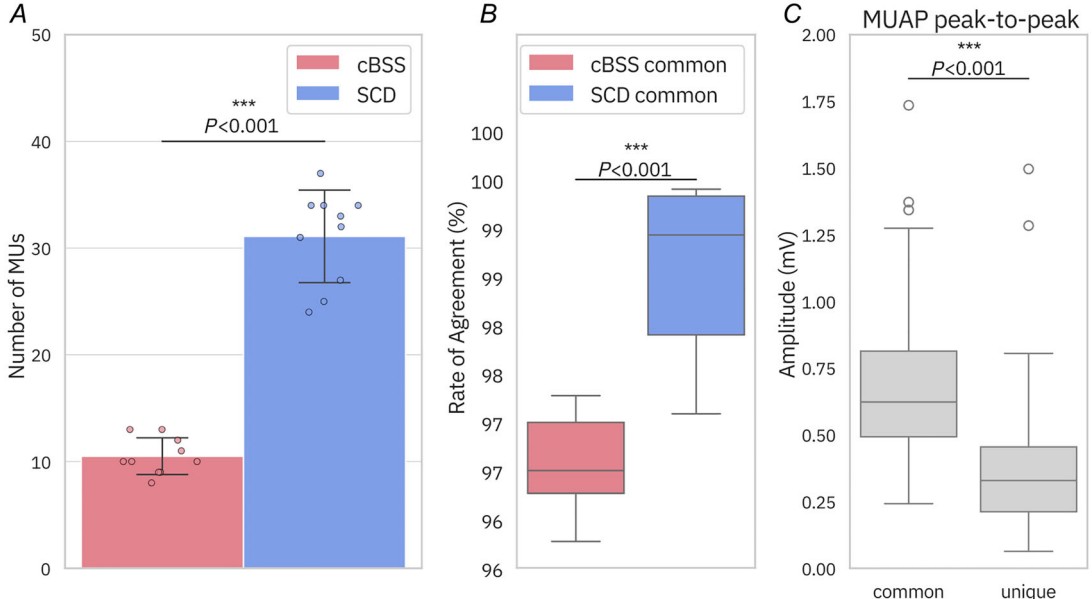

**Figure 6. Effect of ballistic task on the number of motor units found and the RoA with the ground truth**
*A*, number of motor units found for SCD and cBSS. Bars represent the mean of the bootstrapped samples, while error bars reflect the standard deviation across these bootstrapping iterations. Individual data points from each bootstrapping iteration are overlaid on the bar chart. SCD decomposed significantly more motor units than cBSS ($P < 0.001$). *B*, distribution of the RoA between decomposed motor units and their simulated ground truth for the motor units commonly detected by SCD and cBSS. *C*, distributions of the peak-to-peak MUAP amplitudes for motor units common to both SCD and cBSS, and those uniquely identified by SCD. cBSS, convolutive BSS; MUAP, motor unit action potential; RoA, rate of agreement; SCD, Swarm-Contrastive Decomposition. [Colour figure can be viewed at wileyonlinelibrary.com]

(Fig. 3*B*). This is because sources requiring higher or lower exponents would not be detected with a fixed $e = 2.4$, preventing the algorithm from removing their contributions and converging to alternative sources. Moreover, the median exponent varied depending on the dataset, making it impractical to predefine an exponent value in advance; this determination would only be feasible *post hoc*, following decomposition using the swarm update. By incrementally removing the contributions of higher amplitude MUAPs, SCD was able to converge to smaller, less prominent MUAPs that would otherwise remain undetected. Importantly, the peel-off approach did not

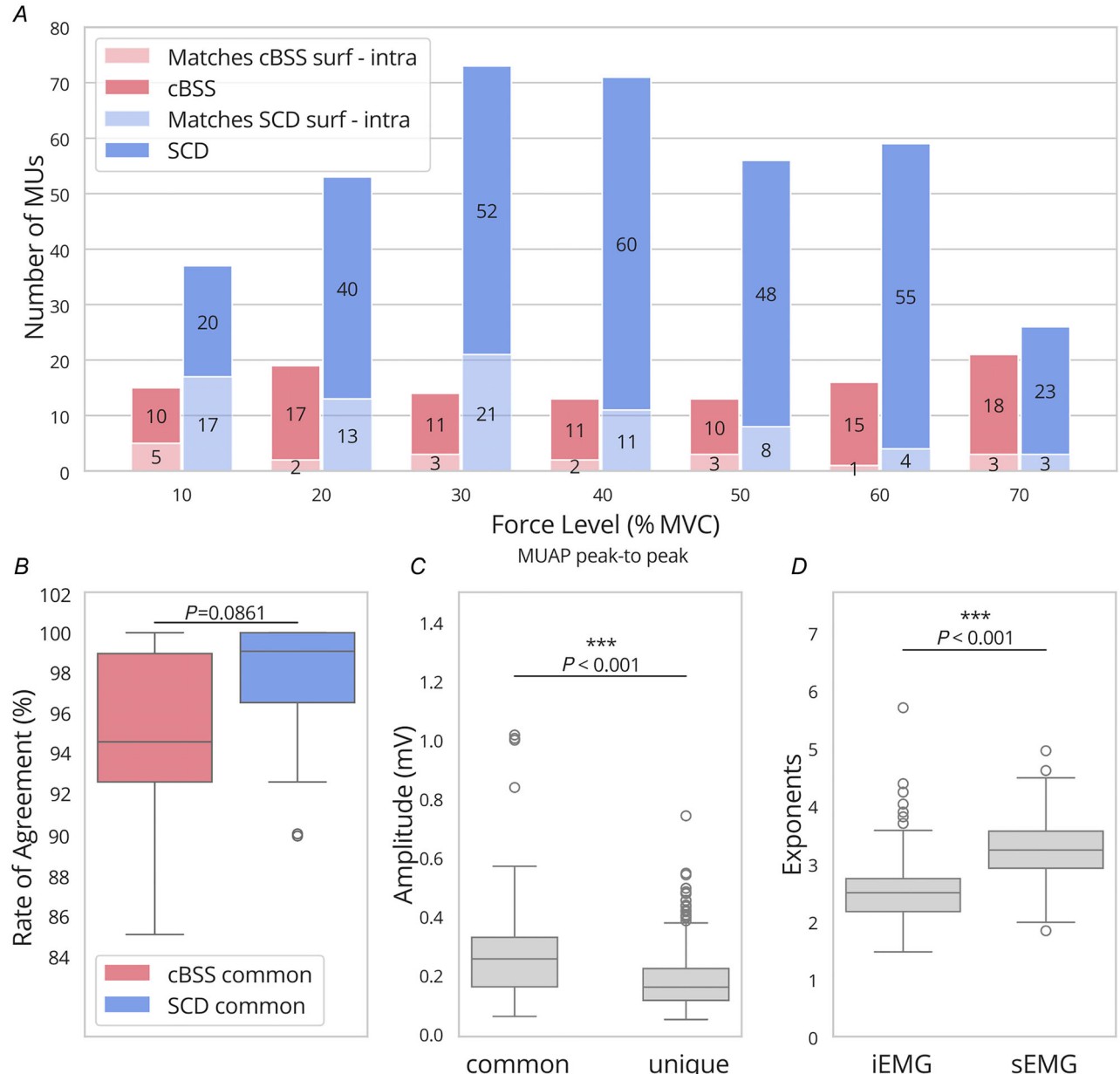

**Figure 7. Effect of the decomposition algorithm used on experimental data**
*A*, number of motor units found against the force level for SCD and cBSS. The number of matched motor units between the intramuscular and the surface recordings is reported in a lighter shade (light pink for cBSS and light blue for SCD). The results from the two subjects are pooled together. *B*, distribution of the RoA for the motor units that matched with the intramuscular recordings and common between cBSS and SCD. *C*, distribution of the MUAP peak-to-peak amplitudes for the motor units commonly identified by cBSS and SCD, and those uniquely found by SCD. *D*, distribution of the exponents for the intramuscular and the surface recordings. cBSS, convolutive BSS; MUAP, motor unit action potential; RoA, rate of agreement; SCD, Swarm-Contrastive Decomposition. [Colour figure can be viewed at wileyonlinelibrary.com]

compromise performance; in all analyses, the RoA for SCD consistently surpassed that of cBSS, which employs a deflation method.

Overall, the simulations indicated high performance of the proposed method in terms of motor unit yield and accuracy in decomposition. We further validated SCD experimentally using a two-source approach by comparing HD-iEMG and HD-sEMG recordings (Farina, Merletti et al., 2014, Farina, Negro et al., 2014; Mambrito & de Luca, 1984). Matching the decomposition results between these two modalities was crucial for validating the proposed algorithm. SCD decomposed significantly more motor units than cBSS from experimental signals, with a higher number of matched motor units extracted from the HD-iEMG recordings ($41.6 \pm 12.1$ and $12.0 \pm 5.3$ *vs.* $13.7 \pm 3.1$ and $2.1 \pm 0.7$, $P < 0.001$).

Interestingly, we observed that the distribution of exponents required in the contrast function had a greater mean value for HD-sEMG signals than for HD-iEMG signals (Fig. 7D). This suggests the need for a higher exponent when dealing with sources that are smaller or more similar to one another, a characteristic often seen with non-invasive signals.

Overall, the new approach significantly increased the number of detected motor units across all conditions without compromising accuracy, which actually improved. The yield varied widely depending on the conditions, with increases ranging from ∼50% to over 300%, but yet an increase was achieved consistently in all cases, across a broad range of scenarios. This consistent increase implies we can now achieve better results in conditions that previously yielded borderline unit detections. Future work could explore how adjusting certain hyperparameters used in SCD might further enhance decomposition performance. For instance, lowering the silhouette threshold or adapting it dynamically during decomposition could potentially yield better results and recover sources that might otherwise be refined during manual editing.

Also, future work could investigate whether all six swarm particles are essential for effectively capturing non-linearities or if computational time can be reduced by using fewer exponents. Nevertheless, since SCD performs six optimizations in parallel and identifies more than twice as many motor units as cBSS, we believe the additional computational time is justified and does not pose a significant barrier to practical application.

The method is particularly useful in conditions where the number of successfully decoded units is typically small. For instance, in cases where the MUAPs are highly similar – such as when recording from deep muscles or from muscles covered by a large layer

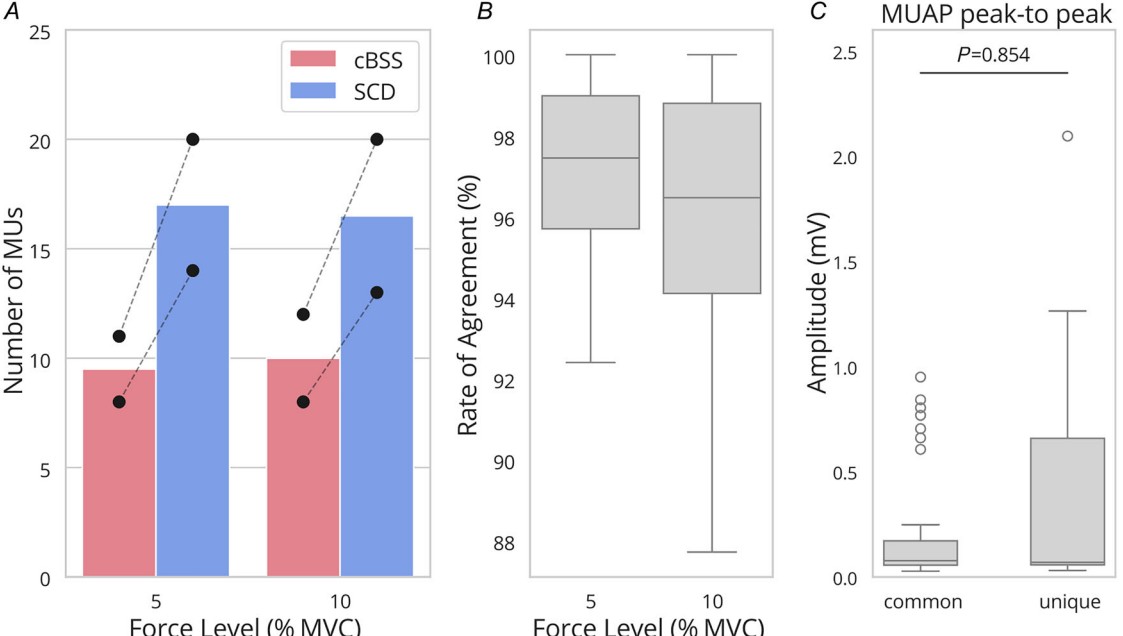

**Figure 8. Effect of the decomposition algorithm used on experimental data**
*A*, mean number of motor units found against the force level for SCD and cBSS across the two participants. Black circles represent the value for each subject. Dashed lines connect the subjects across force levels. *B*, distribution of the RoA for the motor units that matched between cBSS and SCD. *C*, distribution of the MUAP peak-to-peak amplitudes for the motor units commonly identified by cBSS and SCD, and those uniquely found by SCD. cBSS, convolutive BSS; MUAP, motor unit action potential; RoA, rate of agreement; SCD, Swarm-Contrastive Decomposition. [Colour figure can be viewed at wileyonlinelibrary.com]

of subcutaneous fat –discriminating between units becomes a major challenge. In these scenarios, standard decomposition techniques may fail to distinguish MUAPs, often leading to failure in the decomposition. By dynamically optimising the separation of sources, SCD is a step forward to mitigate this problem, maximising the contrast between similar MUAPs and ensuring a higher yield of distinct motor units even in such complex scenarios. An example of such conditions is the decoding of motor units from female individuals. The proposed method demonstrated superior motor unit yield, as highlighted earlier. This suggests that the technique is particularly effective in addressing physiological differences, enhancing the overall accuracy and applicability of motor unit detection across diverse populations. However, due to the limited number of experimental data presented, these findings should be validated through further analysis and additional research.

We presented representative results from female individuals for both the TA muscle, typically known for successful decomposition, and the forearm muscles, where decomposition tends to be less effective. The proposed method consistently improved the yield of decomposed motor units with respect to the state-of-the-art cBSS in these subjects for both muscle groups. However, the limited number of participants and experimental conditions precluded a statistical analysis, highlighting the need for additional data to evaluate the significance of these findings.

In most conditions, in female individuals, we could enhance the number of decomposed units by around 50%. While this does not match the numbers observed in males, it represents a substantial improvement in many cases. In some instances, adding five to six reliably extracted motor units, as shown in this study, can have a significant impact on the physiological interpretations.

Increasing the number of concurrently decoded motor units is critical for advancing our understanding of the neural control of movement. For example, a larger pool of concurrently sampled motor units provides more comprehensive information on the distribution of common synaptic inputs to motor neuron populations (Farina, Merletti et al., 2014, Farina, Negro et al., 2014). This is particularly valuable for characterising how motor neurons are grouped to generate force and for predicting the net muscle force generated by the population of active units (Caillet et al., 2023). Indeed, the association between estimated neural drive to muscle and muscle force output becomes stronger and more reliable as more motor units are included in the cumulative analysis.

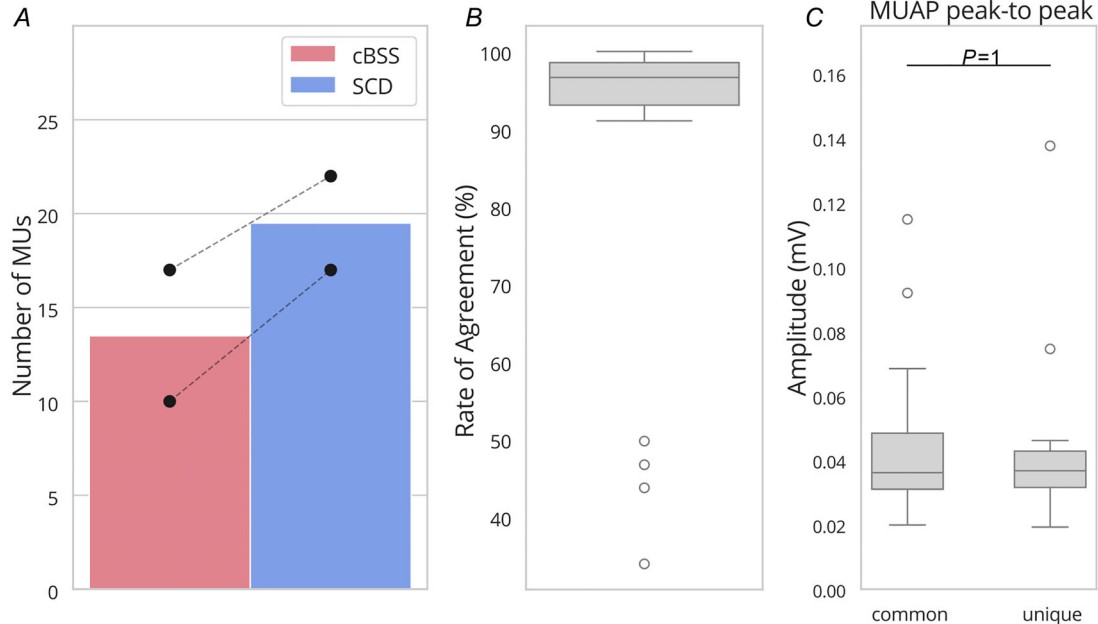

**Figure 9. Effect of the decomposition algorithm used on experimental data recorded at the forearm of two female participants**
*A*, mean number of motor units found for SCD and cBSS across the two participants. Black circles represent the value for each subject. Dashed lines connect the subjects across force levels. *B*, distribution of the RoA for the motor units that matched between cBSS and SCD. *C*, distribution of the MUAP peak-to-peak amplitudes for the motor units commonly identified by cBSS and SCD, and those uniquely found by SCD. cBSS, convolutive BSS; MUAP, motor unit action potential; RoA, rate of agreement; SCD, Swarm-Contrastive Decomposition. [Colour figure can be viewed at wileyonlinelibrary.com]

The expansion in motor unit sample size not only enhances the precision of neuromuscular research but also opens up new possibilities for clinical and assistive applications, such as refined neural control strategies in prosthetics and advanced neural interfaces for rehabilitation (Barsakcioglu et al., 2020; Farina et al., 2017; Gogeascoechea et al., 2020; Tanzarella et al., 2023). In these applications, adapting SCD for real-time, online decomposition is a key future direction.

However, SCD has been validated exclusively in scenarios where the muscle geometry remains static. Its current framework does not account for dynamic changes in muscle fibre length or significant shifts in firing patterns. Consequently, SCD's performance is likely to degrade in more complex conditions, such as during dynamic muscle contractions where fibre lengths or recruitment strategies change over time, causing non-stationary changes in the shapes of the MUAPs. This limitation highlights the need for further development to adapt the algorithm for non-stationary conditions, where changes in muscle architecture and firing behaviour could substantially impact decomposition accuracy.

## Conclusions

We have demonstrated the potential of SCD as a promising benchmark for HD-sEMG decomposition, as supported by strong simulation results and experimental case studies under different challenging conditions. By dynamically adapting its contrast function and adopting a unique peel-off strategy, SCD consistently outperforms traditional methods, particularly in detecting small and deep motor units. These findings, validated through simulations and experimental data, pave the way for SCD's use as a new tool for the study of the neural control of movement as well as for applications ranging from clinical diagnostics to advanced human–machine interfaces. Its ability to resolve finer differences between MUAPs marks a significant leap forward in capturing the full complexity of neuromuscular activity.

The code used in this study is available at https://github.com/AgneGris/swarm-contrastive-decomposition.

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

## Additional information

### Data availability statement

The data that support the findings of this study are available from the corresponding author (D.F.) upon reasonable request.

## Competing interests

None declared.

## Author contributions

A.G.: Conceptualisation, Investigation, Analysis, Writing – review & editing; I.M.G.: Conceptualisation, Investigation, Analysis, Writing – review & editing; A.K.C.: Conceptualisation, Writing – review & editing; S.M.: Conceptualisation, Writing – review & editing; J.I.P.: Conceptualisation, Writing – review & editing; D.F.: Conceptualisation, Investigation, Resources, Writing – review & editing. All authors have approved the final version of the manuscript and agreed to be accountable for all aspects of the work. All persons designated as authors qualify for authorship, and all those who qualify for authorship are listed.

## Funding

A.G. was supported in part by UK Research and Innovation (UKRI Centre for Doctoral Training in AI for Healthcare grant number EP/S023283/1) and in part by Huawei Technologies Research & Development (UK) Ltd. I.M.G. was supported by the EPSRC Doctoral Prize Fellowship. S.M. was supported by HybridNeuro (HORIZON-WIDERA-2021-ACCESS-03 - 101079392). J.I.P. was supported by project ECHOES (ERC Starting Grant 101077693), by a Consolidación Investigadora grant (CNS2022-135366) funded by MCIN/AEI/ 10.13039/501100011033 and UE's NextGenerationEU/PRTR funds. D.F. was supported by NISNEM (EPSRC EP/T020970/1).

## Acknowledgements

We would like to thank the participants who took part in this study. We are also grateful to Dr Alejandro Pascual Valdunciel for providing part of the data utilised in this study. These data were collected at Imperial College London and were compliant with the ethical guidelines set by Imperial College London (ICREC Project ID 19IC5640).

## Keywords

decomposition, motor control, motor units

## Supporting information

Additional supporting information can be found online in the Supporting Information section at the end of the HTML view of the article. Supporting information files available:

**Peer Review History**

