## [Peer Review History · The Journal of Physiology]

Unlocking the Full Potential of High-Density Surface EMG: Novel Non-Invasive High-Yield Motor Unit Decomposition

Agnese Grison, Irene Mendez Guerra, Alexander Kenneth Clarke, Silvia Muceli, Jaime Ibanez Pereda, and Dario Farina
DOI: 10.1113/JP287913

Corresponding author(s): Dario Farina (d.farina@imperial.ac.uk)

Review Timeline:

Submission Date:	22-Oct-2024
Editorial Decision:	09-Dec-2024
Revision Received:	10-Feb-2025
Accepted:	26-Feb-2025

Senior Editor: Richard Carson

Reviewing Editor: Madeleine Lowery

Transaction Report:

Dear Dr Farina,

Re: JP-TFP-2024-287913 "Unlocking the Full Potential of High-Density Surface EMG: Novel Non-Invasive High-Yield Motor Unit Decomposition" by Agnese Grison, Irene Mendez Guerra, Alexander Kenneth Clarke, Silvia Muceli, Jaime Ibanez Pereda, and Dario Farina

Thank you for submitting your manuscript to The Journal of Physiology. It has been assessed by a Reviewing Editor and by 2 expert referees and we are pleased to tell you that it is potentially acceptable for publication following satisfactory major revision.

LANGUAGE EDITING AND SUPPORT FOR PUBLICATION: If you would like help with English language editing, or other article preparation support, Wiley Editing Services offers expert help, including English Language Editing, as well as translation, manuscript formatting, and figure formatting at www.wileyauthors.com/eoo/preparation. You can also find resources for Preparing Your Article for general guidance about writing and preparing your manuscript at www.wileyauthors.com/eoo/prepresources.

REVISION CHECKLIST:

We look forward to receiving your revised submission.

Yours sincerely,

Richard Carson
Senior Editor
The Journal of Physiology

REQUIRED ITEMS

- Include a Key Points list in the article itself, before the Abstract.
- Author photo and profile. First or joint first authors are asked to provide a short biography (no more than 100 words for one author or 150 words in total for joint first authors) and a portrait photograph. These should be uploaded and clearly labelled together in a Word document with the revised version of the manuscript. See Information for Authors for further details.
- Your manuscript must include a complete Additional Information section, including competing interests; funding; author contributions and acknowledgements.
- The Journal of Physiology funds authors of provisionally accepted papers to use the premium BioRender site to create high resolution schematic figures. Follow this link and enter your details and the manuscript number to create and download figures. Upload these as the figure files for your revised submission. If you choose not to take up this offer, we require figures to be of similar quality and resolution. If you are opting out of this service to authors, state this in the Comments section on the Detailed Information page of the submission form. The link provided should only be used for the purposes of this submission. Authors will be charged for figures created on this premium BioRender account if they are not related to this manuscript submission.
- Please upload separate high-quality figure files via the submission form.
- Please ensure that the Article File you upload is a Word file.

- Papers must comply with the Statistics Policy: https://jp.msubmit.net/cgi-bin/main.plex?form_type=display_requirements#statistics.

In summary:

- If $n \leq 30$, all data points must be plotted in the figure in a way that reveals their range and distribution. A bar graph with data points overlaid, a box and whisker plot or a violin plot (preferably with data points included) are acceptable formats.
- If $n > 30$, then the entire raw dataset must be made available either as supporting information, or hosted on a not-for-profit repository, e.g. FigShare, with access details provided in the manuscript.
- 'n' clearly defined (e.g. x cells from y slices in z animals) in the Methods. Authors should be mindful of pseudoreplication.
- All relevant 'n' values must be clearly stated in the main text, figures and tables.
- The most appropriate summary statistic (e.g. mean or median and standard deviation) must be used. Standard Error of the Mean (SEM) alone is not permitted.
- Exact p values must be stated. Authors must not use 'greater than' or 'less than'. Exact p values must be stated to three significant figures even when 'no statistical significance' is claimed.

- Please include an Abstract Figure file, as well as the Figure Legend text within the main article file. The Abstract Figure is a piece of artwork designed to give readers an immediate understanding of the research and should summarise the main conclusions. If possible, the image should be easily 'readable' from left to right or top to bottom. It should show the physiological relevance of the manuscript so readers can assess the importance and content of its findings. Abstract Figures should not merely recapitulate other figures in the manuscript. Please try to keep the diagram as simple as possible and without superfluous information that may distract from the main conclusion(s). Abstract Figures must be provided by authors no later than the revised manuscript stage and should be uploaded as a separate file during online submission labelled as File Type 'Abstract Figure'. Please also ensure that you include the figure legend in the main article file. All Abstract Figures should be created using BioRender. Authors should use The Journal's premium BioRender account to export high-resolution images. Details on how to use and access the premium account are included as part of this email.

Reviewing Editor:

Comments to ensure the paper complies with the Statistics Policy:

The authors are reminded to ensure that the paper complies with the Journal of Physiology Statistics policy, (i.e. data points on graphs if $n < 30$, data given in the article or as supporting information if $n > 30$, SD used (not SEM), precise p values given (no $<$ or $>$).

Comments to the authors:

This study presents a new method for decomposition of high-density surface EMG signals. The results demonstrate improvements in resolving decomposition of complex signals reviewers. The reviewers agree that the results are likely to be of widespread interest to those working in motor unit and high-density EMG decomposition. They have, however, raised a number of important points that need to be addressed before the manuscript is ready for publication.

Senior Editor:

Comments to ensure the paper complies with the Statistics Policy:

Please ensure that in all figure legends, the basis upon which measures of dispersion represented as error bars have been calculated, is stated clearly. A similar point applies to the main body of the text for cases in which statements such as "13.9{plus minus}2.7" appear.

In this regard, note that it is Journal policy that standard deviations should be used, unless a compelling justification for an alternative measure of dispersion is given.

Comments to the Author:

As highlighted by Referee #1, it is essential that a high degree of circumspection accompanies reports of experimental studies based on two participants. While it can be appreciated that the primary intent was the comparison of different decomposition methods, there is necessarily a requirement for cautious interpretation.

Referee #1:

General Comments:

1) Overall, this report describes a method that seems very interesting and useful to the research community.

2) The biggest weaknesses that I found were that the experimental studies had very small sample size (N=2 for three distinct tests), yet the authors made several claims of performance differences in the experimental work when there was not a statistically significant difference in results. Likely as a result of the small sample size, many of your reported comparisons either did not have accompanying statistical tests or the test was not significant. For example, I did not find any statically significant differences reported in the sex difference experiment (only one statistical result is reported). The two-source validation did report a statistical difference in the number of identified MUs, and this two-source result is likely your strongest experimental evidence, anyway. But, this results is still only from two subjects. Nonetheless, the Discussion and Conclusions seem to group all of the experimental work with the simulation work and make sweeping claims as to the better performance of your method. In general, a Discussion section should not claim that a result is "better" unless a statistical difference is found. Your experimental results seem encouraging, but too small and limited in number and type of test to be overly convincing.

That said, your work provides a LOT of evidence that you may have a new and better decomposition method. I just think that you need to keep your claims limited to the limitations of your results. I would encourage that you re-cast each of your three experimental trials as small "pilot" experiments. If you can align your conclusions to the evidence that you have, I believe that you still have a strong contribution. But, I would strongly advocate that the strength of the conclusions be more limited, based on your sample sizes.

3) It would appear that your method still relies on a repeatable shape for a MU and thus may not be applicable to clinical patients with neuropathies/myopathies, muscles that are fatiguing, etc. It may help to make this point early in the manuscript. [You do note after Equation (1) that the shapes are constant, but it may help to point out what this means.]

4) Results Figure 2 b shows a rather small range of exponent values found using your SCD method, with the median around 2.4. On page 11 near the bottom, you compared your adaptive exponent to fixed values of 2 and 3. Assuming that exponent values need not be integer (your model does not comment explicitly on this assumption), Figure 2 b does not seem to provide a convincing argument for adaptive exponents, rather it seems to suggest that an exponent near 2.4 might be best. Thus, can you really conclude that your adaptive exponent selection was better (than fixing the exponent at about 2.4)?

5) Simulation RoA results: There are several cases where you found statistically significant differences, however both SCA and cBSS had performance very close to the maximum of 100%. In this case, even though the results differed statistically, was the strength of the difference important? (Statistical difference vs. statistical strength)

6) Can you provide a rough estimate of the amount of computation required for this algorithm, in general and compared to cBSS? (A single added sentence might help.)

7) The Acknowledgements state that Dr. Valdunciel provided data utilized in this study. Proper human studies permissions should be provided if Dr. Valdunciel provided experimental data from humans. (No such permission is required, of course, for simulation data.)

Specific Comments:

8) Abstract: If possible, it would help to include quantitative performance results in the Abstract. At present, the only quantitative performance description is the second-to-last sentence, and it limits itself to "certain experimental conditions." Can you quantitatively describe the increase in performance? Doing so convinces a reader whether or not to read past the Abstract section.

9) P 2, PP 2, "These recordings...": The high selectivity also has advantages in that it has historically made decomposition possible (prior to more recent source separation methods). And, indwelling EMG can successfully decompose deeper muscle tissue. Would it be more balanced to also mention these issues?

10) Also, a sentence later: The reason for controlled datasets does not seem to follow your logic (high degree of selectivity). There are other issues for the controlled experiments, e.g., movement of the electrodes (during limb movement) can dislodge the electrodes such that they no longer record the same MUs; or high contraction level may produce too many overlapping MUs such that simple template-based decomposition fails. Perhaps it is just the wording in this section, but some of what you are stating, while correct in general, does not seem to flow. Historically, the high selectivity of indwelling recordings has been a "feature and not a bug."

11) After Equations (1) and (2) (perhaps elsewhere?), there should be no paragraph indent prior to "where".

12) For Equations 3 and 4, can you also provide the dimensions? (They might be inferred from the prior equations, but it may be more helpful to give them explicitly.)

13) P 5, SCD initialization of w : What was the standard deviation of the zero-mean normally distributed vectors and were they mutually independent?

14) P 5, near bottom: You note here an assumption of isometric contractions. I would suggest also listing this assumption much earlier in the manuscript. Do you also assume isotonic contractions (so that the MU sources do not "slide" relative to the electrode array)?

15) P 6, line 2: Can you clarify what is meant by "peeled off"? Did you create a template MU, then subtract it from the signal at each MU firing (based on your decomposition)?

16) P 6, bottom: Should "1 a" be "1a"?

17) P 7, line 1: Did you truncate this normal distribution (since rare very slow/fast values are possible; hence if the technique is used enough, such undesirable values WILL occur)?

18) P 7, first paragraph, second-to-last line: Again, a full description of the Gaussian noise is needed (independent? zero-mean?).

19) P 7: Since some of the contractions were NOT isotonic, I would recommend listing for each (most?) condition prior to the "second" analysis that the contractions were isotonic (e.g., change "contraction lasted 30 s" to "isotonic contraction lasted 30 s").

20) P 8, top: In first paragraph, you refer to "Figure 1 a" then later in the paragraph as "Figure 1, a". Please use journal style consistently.

21) P 8, top paragraph: Please provide more details about the electrode array. What are the electrode contact materials, inter-electrode spacing, number of electrodes per row and column, etc.? In general, it is also expected to list the CMRR of the amplifier (at the power-line frequency) and the input-referred noise.

22) Section 2.4.1: Is this the same array as mentioned in the prior paragraph? If the system has several available arrays, perhaps tell the reader?

23) Section 3.2, p=0.18: Did you perform a single statistical test that included all noise levels, or is this the lowest p-value? If lowest, it would usually be written as $p \geq 0.18$.

24) What are the y-axis units for Figure 8 c and Figure 9 c?

25) P 20, bottom, "... the RoA ... was greater for SCD (Figure 7, b)". I believe that your statistical test for this difference was not significant ($p=0.08$). So, I do not believe that you are justified in making this statement.

26) P 23, "This suggests ... diverse populations.": Even with the qualifier of "suggests", this statement seems rather strong since your subject population corresponding to this statement was $N=2$. I would recommend even more qualification/limiting of this statement.

27) P 23, "The proposed method ... both muscle groups.": The only statistical tests that I could find in the sex difference results was "not significant." So, this statement does not seem to be supported.

28) Conclusions, first sentence: While your simulation results are quite convincing, your experimental results are from a small data set wherein the primary statistical differences were in the number of units identified. Your contractions are isometric and mostly isotonic. So, this statement seems too strong. Your simulation results seem like an excellent argument, but additional experimental results would be welcomed and more convincing.

29) Conclusions: I would not classify the experimental work herein as "comprehensive." You have three conditions, each with two subjects. Most convincing experimentally are the two-source results. But, still $N=2$.

30) It is wonderful that you are sharing the code!

31) You used the same silhouette value as Negro et al. 2016. Since your decomposition performance (RoA) is so high, I wonder if your technique could support a lower silhouette value, perhaps retaining high RoA with the added benefit of identifying even more MUs? (Not asking you to test this value as additional work, but wonder if an appropriate statement might appear in the Discussion---as a form of future work, perhaps.)

Referee #2:

Please see attached file [JP-TFP-2024-287913_Referee 2 Review Attachment 1.pdf]

END OF COMMENTS

General Comments:

This manuscript describes and evaluates a new method, SCD, for extracting MUPTs from HD_EMG signals using particle-swarm optimization to iteratively select optimal contrast functions. The methods section is not clearly written and would be difficult for most JP readers to understand. It should be revised to present the SCD algorithm more clearly and from more conceptual basis.

Specific Comments:

What is the difference between a separation function and a contrast function?

Applying HD-EMG methods to the quadriceps muscle can be very difficult. Why was it not studied?

“ ... triggers a motor unit action potential (MUAP) in the muscle fibers ... ”
MUAPs are not triggered in muscle fibers?

“EMG decomposition aims to separate these two components ...”
EMG decomposition algorithms separate MN discharge times from MUPs?

The concepts of cBSS are not described in a way that most JP readers will easily understand. The reasons for the successes of SCD will likewise not be easily understood.

Perhaps a brief conceptual description of ICA, how it works and why and how it can be applied to decompose HD-EMG signals would make the subject matter more accessible to the reader. In addition, perhaps a brief conceptual description of what a contrast function is and how one is used in the decomposition of HD-EMG signals would be useful.

For the Grison et al., 2024 reference is the authors list correct?

“ Similar to the Gaussianity property ...” ?

The importance of finding sources that are maximally non-Gaussian is not clear.
The connection between sparseness, skewness, kurtosis and non-Gaussianity is not clear.
Hyvarinen & Oja, 2000 does not mention sparseness?
Why is a more sparse source considered to be a more correct source?
What is y ? Is it a scalar, representing one channel, or a vector, representing all channels?

How is the contrast function for each source determined and used?
It is not clear how the use of a different “optimal” contrast function for each source is beneficial.
“ ... the loss $G(s)$..” should be “ ... the last $G(s)$...”?

How is failure of a $G(s)$ to provide improvement determined?

“After each ICA step ...” What was the output of each ICA step?

“ ... two-class k -medoid clustering to identify potential motor unit spikes.”
What was the input to the clustering algorithm? How are the discovered clusters used?

What is are “potential motor unit spikes”?

“ ... using a fitness function based on the coefficient of variation of the interspike intervals ...”
When studying unknown MN firing patterns, why is it useful to assume that their patterns will be

maximally regular?

“... the regularity of motor unit firing in isometric contractions.”

What about non-isotonic force contractions?

“After each ICA update ...” How was the ICA updated? What about the ICA was updated?

20 iterations or ten updates which is it?

Was each source iteratively estimated one at a time, via the use of its optimal $G(s)$, followed by its “peeling-off”?

What happens if a MUPT does not pass the silhouette test or meet the COV and firing rate criteria?

Is it still peeled off?

“ablation” does not seem to be the best term to use here?

“The innervated muscle fibers were randomly and uniformly distributed throughout the muscle volume ...”

Were MU territories used?

“... decreases in female individuals”

Is this not a bigger problem when studying the quadriceps then the TA?

If so, why were quadriceps studies not included?

“...such as when recording from deep muscles or from muscles covered by a large layer of subcutaneous fat ...”

Again, why were quadriceps studies not included? The TA is quite a superficial muscle by comparison.

Reviewing Editor:

=====

Comments to ensure the paper complies with the Statistics Policy:

The authors are reminded to ensure that the paper complies with the Journal of Physiology Statistics policy,(i.e. data points on graphs if $n < 30$, data given in the article or as supporting information if $n > 30$, SD used (not SEM), precise p values given (no $<$ or $>$).

Thank you, we have made the necessary changes to the manuscript to comply with the journal Statistics policies. These include 1) presenting all data points when $n < 30$ (for $n > 30$, the data can be requested from the corresponding author as detailed in the Data Availability statement); 2) giving precise p values (apart from when $p < 0.001$ as per Journal standards); 3) presenting the mean and standard deviation for the Rate of Agreement instead of the median and interquartile range.

Comments to the authors:

This study presents a new method for decomposition of high-density surface EMG signals. The results demonstrate improvements in resolving decomposition of complex signals reviewers. The reviewers agree that the results are likely to be of widespread interest to those working in motor unit and high-density EMG decomposition. They have, however, raised a number of important points that need to be addressed before the manuscript is ready for publication.

REPLY:

We sincerely appreciate the constructive feedback provided by the reviewers. Each comment has been carefully addressed in the point-by-point replies below, and the manuscript has been revised accordingly.

The key revisions include:

1. A new ablation analysis added to the *Results* section.
2. Additional details and clarifications incorporated into the *Methods* section.
3. A more balanced discussion, with claims adjusted to reflect the limitations of the small sample sizes.
4. Conformed the manuscript with the Journal standards regarding statistical analysis, reporting and data visualisation.

We believe these revisions have strengthened our manuscript, and we are grateful to the reviewers for their insightful contributions, which have helped improve the clarity and rigor of our work.

Senior Editor:

=====

Comments to ensure the paper complies with the Statistics Policy:

Please ensure that in all figure legends, the basis upon which measures of dispersion represented as error bars have been calculated, is stated clearly. A similar point applies to the main body of the text for cases in which statements such as "13.9{plus minus}2.7" appear.

In this regard, note that it is Journal policy that standard deviations should be used, unless a compelling justification for an alternative measure of dispersion is given.

Thank you. We have modified the text of the manuscript to report the mean and the standard deviation for all analysis. We have also ensured that all figure legends report what the error bars represent.

Comments to the Author:

As highlighted by Referee #1, it is essential that a high degree of circumspection accompanies reports of experimental studies based on two participants. While it can be appreciated that the primary intent was the comparison of different decomposition methods, there is necessarily a requirement for cautious interpretation.

REPLY:

We thank the editor for their feedback. We have provided our point-to-point replies below.

Referee #1:

=====

General Comments:

1) Overall, this report describes a method that seems very interesting and useful to the research community.

REPLY:

We thank the reviewer for their appreciation of our work.

2) The biggest weaknesses that I found were that the experimental studies had very small sample size (N=2 for three distinct tests), yet the authors made several claims of performance differences in the experimental work when there was not a statistically significant difference in results. Likely as a result of the small sample size, many of your reported comparisons either did not have accompanying statistical tests or the test was not significant. For example, I did not find any statically significant

differences reported in the sex difference experiment (only one statistical result is reported). The two-source validation did report a statistical difference in the number of identified MUs, and this two-source result is likely your strongest experimental evidence, anyway. But, this results is still only from two subjects. Nonetheless, the Discussion and Conclusions seem to group all of the experimental work with the simulation work and make sweeping claims as to the better performance of your method. In general, a Discussion section should not claim that a result is "better" unless a statistical difference is found. Your experimental results seem encouraging, but too small and limited in number and type of test to be overly convincing.

That said, your work provides a LOT of evidence that you may have a new and better decomposition method. I just think that you need to keep your claims limited to the limitations of your results. I would encourage that you re-cast each of your three experimental trials as small "pilot" experiments. If you can align your conclusions to the evidence that you have, I believe that you still have a strong contribution. But, I would strongly advocate that the strength of the conclusions be more limited, based on your sample sizes.

REPLY:

We appreciate the reviewer's comments and acknowledge the concerns regarding the small sample size in our experimental studies. We agree that the limited number of subjects constrained our ability to conduct statistical analyses in some cases (where $N=2$). This is commented in the limitations part of the manuscript. However, we think that it is important to note that the two-source validation, with comparison with intramuscular signals, remains a critical component of the study, providing the most direct means of validating the decomposition of experimentally recorded surface EMG signals. The small subject sample is due to the relative complexity in this validation approach. On the other hand, this approach provides a very strong indication of accuracy compared across algorithms. Without this approach, it would have not been possible to provide a detailed comparison among algorithms experimentally.

In response to the reviewer's suggestion, we have explicitly redefined our experimental studies as *Pilot 1*, *Pilot 2*, and *Pilot 3* in the *Methods* and *Results* sections. Additionally, we have revised the *Discussion* and *Conclusion* sections to clearly indicate the limitations of our study and to ensure that our claims align with the scope of our experimental evidence. We believe these refinements improve the clarity and accuracy of our conclusions while maintaining the significance of our contributions.

3) It would appear that your method still relies on a repeatable shape for a MU and thus may not be applicable to clinical patients with neuropathies/myopathies, muscles that are fatiguing, etc. It may help to make this point early in the manuscript. [You do note after Equation (1) that the shapes are constant, but it may help to point out what this means.]

REPLY:

Thank you for your insightful comment. Indeed, a fundamental assumption of our model is that MUAPs remain similar over time. While this assumption simplifies the decomposition process, it also imposes limitations on the conditions in which our method can be effectively applied.

To address this issue, we have clarified this point in the Methods section:

"As this method assumes the repeatability of motor unit action potential shapes, it is not applicable in scenarios where the waveform shapes exhibit non-stationary variations, such as those caused by electrode displacement, fatigue, joint movement, or certain clinical conditions."

Moreover, we have emphasised this limitation in the *Discussion*, indicating that SCD does not account for non-stationary changes in the MUAP shapes.

4) Results Figure 2 b shows a rather small range of exponent values found using your SCD method, with the median around 2.4. On page 11 near the bottom, you compared your adaptive exponent to fixed values of 2 and 3. Assuming that exponent values need not be integer (your model does not comment explicitly on this assumption), Figure 2 b does not seem to provide a convincing argument for adaptive exponents, rather it seems to suggest that an exponent near 2.4 might be best. Thus, can you really conclude that your adaptive exponent selection was better (than fixing the exponent at about 2.4)?

REPLY:

We thank the reviewer for their insightful comment. Indeed, the exponent values in our model do not need to be integers, and fixing the exponent to $e=2.4$ is a practical consideration. In response to this suggestion, we conducted an additional analysis using a fixed exponent of $e=2.4$ and have incorporated the results into the updated *Figure 3b*.

To further evaluate the impact of adaptive exponent selection, we performed a two-way repeated measures ANOVA, with the number of decomposed motor units as the dependent variable and both MVC level and exponent as independent variables. The results indicate that the adaptive exponent selection identified significantly more motor units than any fixed exponent choice ($p<0.001$).

We have added this information to the *Results* section and expanded the *Discussion* to address the implications of using a fixed exponent at the median value:

"Interestingly, we showed that fixing the exponent to the median value of the adaptively determined exponents during the swarm update yielded significantly fewer decomposed motor units compared to allowing the exponent to adapt (Figure 3b). This is because sources requiring higher or lower exponents would not be detected with a fixed $e=2.4$, preventing the algorithm from effectively removing their contributions and converging to alternative sources. Moreover, the median exponent varied depending on the dataset, making it impractical to predefine a fixed exponent value in advance; this determination would only be feasible post hoc, following decomposition using the swarm update."

We appreciate the reviewer's suggestion, as it has helped us strengthen our argument and provide a clearer justification for the benefits of adaptive exponent selection.

5) Simulation RoA results: There are several cases where you found statistically significant differences, however both SCA and cBSS had performance very close to the maximum of 100%. In this case, even though the results differed statistically, was

the strength of the difference important? (Statistical difference vs. statistical strength)

REPLY:

Thank you for your comment. To address this point, we have incorporated a measure of effect size for the differences in *RoA* in *Section 3.1*. Specifically, we have added the following statement:

"The effect size, as measured by a Rank-Biserial Correlation of 0.99, indicates a substantial and practically meaningful improvement in RoA."

While the absolute *RoA* values for *SCA* and *cBSS* appear close to the maximum (100%), the underlying distributions differ significantly, as illustrated in *Figure 2c*. This distinction is captured by the effect size, which confirms that the observed statistical difference corresponds to a meaningful improvement in performance.

6) Can you provide a rough estimate of the amount of computation required for this algorithm, in general and compared to cBSS? (A single added sentence might help.)

REPLY:

Thank you for your comment. We have added the following clarification in the *Methods* section:

"The decompositions were run on an Intel(R) Core(TM) i7-10700K CPU with an Nvidia RTX 3080 GPU." In the Results, we have added "The computational time required to run one decomposition was approximately 2 minutes for cBSS and 12 minutes for SCD."

Given that *SCD* effectively runs six optimizations in parallel and identifies more than twice as many motor units as *cBSS*, we believe that the increased computational time is justified and does not present a significant barrier to its practical use. We have added this reasoning in the *Discussion*.

7) The Acknowledgements state that Dr. Valdunciel provided data utilized in this study. Proper human studies permissions should be provided if Dr. Valdunciel provided experimental data from humans. (No such permission is required, of course, for simulation data.)

REPLY:

Thank you for your comment. We confirm that the data provided by Dr. Valdunciel was collected in the same laboratory at Imperial College London under the same ethical guidelines as the rest of the study. To ensure clarity, we have revised the *Acknowledgments* section to explicitly state:

"This data was collected at Imperial College London and was compliant with the ethical guidelines set by Imperial College London (ICREC Project ID 19IC5640)."

Additionally, we have updated the *Methods* section describing the experimental data by replacing *"The experiments"* with *"All reported experiments"* to reinforce that all human data

adhered to the same ethical standards.

Specific Comments:

8) Abstract: If possible, it would help to include quantitative performance results in the Abstract. At present, the only quantitative performance description is the second-to-last sentence, and it limits itself to "certain experimental conditions." Can you quantitatively describe the increase in performance? Doing so convinces a reader whether or not to read past the Abstract section.

REPLY:

Thank you for your comment. We have added quantitative information in the abstract to describe the increase in performance.

9) P 2, PP 2, "These recordings...": The high selectivity also has advantages in that it has historically made decomposition possible (prior to more recent source separation methods). And, indwelling EMG can successfully decompose deeper muscle tissue. Would it be more balanced to also mention these issues?

REPLY:

Thank you for your comment. We have revised the paragraph to ensure a more balanced discussion. The revised text highlights these important contributions while maintaining our perspective on the scalability and generalizability challenges associated with intramuscular EMG.

10) Also, a sentence later: The reason for controlled datasets does not seem to follow your logic (high degree of selectivity). There are other issues for the controlled experiments, e.g., movement of the electrodes (during limb movement) can dislodge the electrodes such that they no longer record the same MUs; or high contraction level may produce too many overlapping MUs such that simple template-based decomposition fails. Perhaps it is just the wording in this section, but some of what you are stating, while correct in general, does not seem to flow. Historically, the high selectivity of indwelling recordings has been a "feature and not a bug."

REPLY:

Thank you for your comment. We have clarified that a limitation of intramuscular EMG is its reliance on highly controlled experimental conditions. The revised sentence now states:

"For nearly a century, the primary method for investigating motor units has been through invasive EMG techniques, involving the decomposition of recordings from needle or wire electrodes under highly controlled conditions."

We acknowledge that factors such as high contraction forces and significant limb movements can affect intramuscular recordings, leading to issues such as electrode displacement or excessive MU overlap, which can hinder decomposition. Our intention is to highlight that while the high selectivity of intramuscular EMG was instrumental in enabling

early decomposition methods, this same selectivity presents challenges when aiming for broader population studies, generalization, and scalability.

11) After Equations (1) and (2) (perhaps elsewhere?), there should be no paragraph indent prior to "where".

REPLY:

Thank you for your comment. This has now been fixed throughout the manuscript.

12) For Equations 3 and 4, can you also provide the dimensions? (They might be inferred from the prior equations, but it may be more helpful to give them explicitly.)

REPLY:

Thank you for your comment. We have now added the dimensions of the matrices.

13) P 5, SCD initialization of w: What was the standard deviation of the zero-mean normally distributed vectors and were they mutually independent?

REPLY:

Thank you for your comment. We have incorporated this information into the updated paragraph:

"In SCD, a candidate separation vector was randomly initialised from a zero-mean normal distribution with a standard deviation of 1. The candidate vector was then repeated for the number of initialised particles to produce the initial separation vectors w . On each step, an ICA run was conducted on each separation vector independently for a maximum of 1000 iterations, with an early termination criterion applied if, after 20 iterations, $G_SCD(s)$ ceased to increase."

14) P 5, near bottom: You note here an assumption of isometric contractions. I would suggest also listing this assumption much earlier in the manuscript. Do you also assume isotonic contractions (so that the MU sources do not "slide" relative to the electrode array)?

REPLY:

Thank you for your comment. We believe that the paragraph outlining the assumption of MUAP shape repeatability effectively addresses this concern:

"As this method assumes the repeatability of motor unit action potential shapes, it is not applicable in scenarios where the waveform shapes exhibit non-stationary variations, such as due to sliding of the electrodes, fatigue, movement of the joints, or clinical conditions."

Throughout the manuscript, we assume that the length of the muscle does not change, so we assume isometric contractions.

15) P 6, line 2: Can you clarify what is meant by "peeled off"? Did you create a template MU, then subtract it from the signal at each MU firing (based on your decomposition)?

REPLY:

Thank you for your comment. To clarify the meaning of "*peeled off*," we have revised the text as follows:

"If the source passed this evaluation ($silhouette > 0.85$) and had not been identified in previous iterations, the motor unit's contributions were subtracted—i.e., peeled off—from the signal by creating a template motor unit waveform and removing it at each identified motor unit firing time. This subtraction prevented further convergence to the same source in subsequent decomposition iterations."

This revision explicitly describes the subtraction process to ensure clarity for the reader.

16) P 6, bottom: Should "1 a" be "1a"?

REPLY:

Thank you for your comment. We have corrected all Figure references throughout the manuscript.

17) P 7, line 1: Did you truncate this normal distribution (since rare very slow/fast values are possible; hence if the technique is used enough, such undesirable values WILL occur)?

REPLY:

Thank you for your comment. The normal distribution is indeed truncated between 3 and 4.5 m/s to avoid outliers. We have now added this detail in the manuscript.

18) P 7, first paragraph, second-to-last line: Again, a full description of the Gaussian noise is needed (independent? zero-mean?).

REPLY:

Thank you for your comment. The noise was zero-mean white Gaussian noise with standard deviation based on the amplitude of the originally simulated noiseless EMG signals across time and channels across channels. We have clarified this in the *Methods*.

19) P 7: Since some of the contractions were NOT isotonic, I would recommend listing for each (most?) condition prior to the "second" analysis that the contractions were isotonic (e.g., change "contraction lasted 30 s" to "isotonic contraction lasted 30 s").

REPLY:

Thank you for your comment. We have now specified when the contraction was only isometric or isometric and isotonic throughout the manuscript.

20) P 8, top: In first paragraph, you refer to "Figure 1 a" then later in the paragraph as "Figure 1, a". Please use journal style consistently.

REPLY:

Thank you for your comment. We have corrected all references for consistency.

21) P 8, top paragraph: Please provide more details about the electrode array. What are the electrode contact materials, inter-electrode spacing, number of electrodes per row and column, etc.? In general, it is also expected to list the CMRR of the amplifier (at the power-line frequency) and the input-referred noise.

REPLY:

Thank you for your comment. The *Experimental Data* paragraph preceding *Section 2.4.1* provides a general overview of the experimental setups and the tasks performed by the subjects. To address the reviewer's request for more details on the electrode array, we have added the following description in *Section 2.4.1*:

"The electrodes, made of platinum, are arranged on two sides of a wider filament, with each side consisting of 20 electrodes spaced 1 mm apart. The two sides are offset by 0.5 mm (Muceli et al., 2022)."

For additional specifications we refer the reader to the cited reference, which provides a comprehensive technical description of the system.

In addition, we have added the CMRR of the Quattrocento amplifier.

22) Section 2.4.1: Is this the same array as mentioned in the prior paragraph? If the system has several available arrays, perhaps tell the reader?

REPLY:

Thank you for your comment. Only one type of intramuscular electrode was utilised in this study. We have added additional details on the system:

"The electrodes, made of platinum, are arranged on two sides of a wider filament, with each side consisting of 20 electrodes spaced 1 mm apart. The two sides are offset by 0.5 mm (Muceli et al., 2022)."

23) Section 3.2, p=0.18: Did you perform a single statistical test that included all noise levels, or is this the lowest p-value? If lowest, it would usually be written as $p \geq 0.18$.

REPLY:

To ensure a more appropriate statistical analysis given the experimental conditions, we have updated our approach to a two-way repeated measures ANOVA. This analysis accounts for all noise levels, with the number of decomposed units as the dependent variable and both noise level and algorithm type as independent variables.

We have revised the text accordingly to reflect this improved statistical methodology. We appreciate the reviewer's suggestion, which has helped us refine our analysis and ensure the robustness of our results.

24) What are the y-axis units for Figure 8 c and Figure 9 c?

REPLY:

Thank you for noticing this important detail. We have now added the y-label as "Amplitude (mV)".

25) P 20, bottom, "... the RoA ... was greater for SCD (Figure 7, b)". I believe that your statistical test for this difference was not significant (p=0.08). So, I do not believe that you are justified in making this statement.

REPLY:

Thank you for your comment. Indeed, the p-value was not statistically significant, so we removed this statement from the *Discussion*.

26) P 23, "This suggests ... diverse populations.": Even with the qualifier of "suggests", this statement seems rather strong since your subject population corresponding to this statement was N=2. I would recommend even more qualification/limiting of this statement.

REPLY:

To further qualify our statement given the small sample size (N=2), we have added the following clarification:

"However, due to the limited number of experimental studies presented, these findings should be validated through further analysis and additional research."

This revision ensures a more cautious interpretation of our results while acknowledging the need for further validation.

27) P 23, "The proposed method ... both muscle groups.": The only statistical tests that I could find in the sex difference results was "not significant." So, this statement does not seem to be supported.

REPLY:

Thank you for your comment. To ensure that our statement accurately reflects the statistical results, we have added the following clarification:

"However, the limited number of participants and experimental conditions precluded a statistical analysis, highlighting the need for additional data to evaluate the significance of these findings."

This revision acknowledges the limitations of our dataset and avoids making unsupported claims.

28) Conclusions, first sentence: While your simulation results are quite convincing, your experimental results are from a small data set wherein the primary statistical differences were in the number of units identified. Your contractions are isometric and mostly isotonic. So, this statement seems too strong. Your simulation results seem like an excellent argument, but additional experimental results would be welcomed and more convincing.

REPLY:

Thank you for your comment. To better reflect the strengths and limitations of our study, we have revised the first sentence as follows:

" We have demonstrated the potential of SCD as a promising benchmark for HD-sEMG decomposition, as supported by strong simulation results and experimental case studies under different challenging conditions.

29) Conclusions: I would not classify the experimental work herein as "comprehensive." You have three conditions, each with two subjects. Most convincing experimentally are the two-source results. But, still N=2.

REPLY:

Thank you for your comment. We agree, and we have removed the term "comprehensive".

30) It is wonderful that you are sharing the code!

REPLY:

Thank you! We strongly believe in the importance of open science and reproducibility, and we are pleased to share our code to facilitate further research and collaboration in this field.

31) You used the same silhouette value as Negro et al. 2016. Since your decomposition performance (RoA) is so high, I wonder if your technique could support a lower silhouette value, perhaps retaining high RoA with the added benefit of identifying even more MUs? (Not asking you to test this value as additional work, but wonder if an appropriate statement might appear in the Discussion---as a form of future work, perhaps.)

REPLY:

Thank you for your comment. To address this point, we have added the following statement to the *Discussion* as a potential avenue for future work:

"Future work could explore how adjusting certain hyperparameters used in SCD might further enhance decomposition performance. For instance, lowering the silhouette threshold or adapting it dynamically during decomposition could potentially yield better results and recover sources that might otherwise be refined during manual editing."

This addition highlights the possibility of optimizing the silhouette threshold while maintaining high *RoA*, providing a direction for future research.

Referee #2:

General Comments:

This manuscript describes and evaluates a new method, SCD, for extracting MUPTs from HD_EMG signals using particle-swarm optimization to iteratively select optimal contrast functions. The methods section is not clearly written and would be difficult for most JP readers to understand. It should be revised to present the SCD algorithm more clearly and from more conceptual basis.

REPLY:

Specific Comments:

1) What is the difference between a separation function and a contrast function?

REPLY:

Thank you for your comment. The terms *separation function* and *contrast function* are interchangeable, but *contrast function* is the more commonly used terminology. To maintain consistency throughout the manuscript, we have standardized the terminology by using *contrast function*, including in the abstract.

2) Applying HD-EMG methods to the quadriceps muscle can be very difficult. Why was it not studied?

REPLY:

While we acknowledge that applying HD-EMG methods to the quadriceps can be challenging and an interesting avenue for investigation, testing the algorithm on several muscle groups was beyond the scope of this study. However, to facilitate further exploration, we have made our code openly available, allowing researchers to apply the method to different muscle groups, including the quadriceps, in future studies.

We appreciate the reviewer's interest in this aspect and recognize that extending our approach to additional muscles is an important direction for future work.

3) “ ... triggers a motor unit action potential (MUAP) in the muscle fibers ... ” MUAPs are not triggered in muscle fibers?

REPLY:

We have rephrased in:

“Each motor neuron's discharge triggers an action potential in each muscle fiber it innervates (Kandel et al., 2000). The sum of the action potentials of the fibres innervated by a single motor neuron is the MUAP.”

4) “EMG decomposition aims to separate these two components ...” EMG decomposition algorithms separate MN discharge times from MUPs?

REPLY:

Yes, in some respects, EMG decomposition algorithms aim to separate motor unit discharge timings from MUAPs, so that only the discharge times can be further analysed.

5) The concepts of cBSS are not described in a way that most JP readers will easily understand. The reasons for the successes of SCD will likewise not be easily understood. Perhaps a brief conceptual description of ICA, how it works and why and how it can be applied to decompose HD-EMG signals would make the subject matter more accessible to the reader. In addition, perhaps a brief conceptual description of what a contrast function is and how one is used in the decomposition of HD-EMG signals would be useful.

REPLY:

Thank you for your comment. While we agree that a deeper conceptual explanation of cBSS and ICA could enhance accessibility, providing a full methodological background falls outside the scope of this paper. Instead, we have now ensured that key concepts are described more clearly throughout the *Methods* section and have included references to foundational papers for readers seeking a more thorough understanding of ICA, contrast functions, and their application in HD-EMG decomposition.

6) For the Grison et al., 2024 reference is the authors list correct?

REPLY:

Thank you for catching this error. We have corrected the author list by adding the missing part of the surname (*Pereda*) and adjusting *Ibanez* to *Ibañez*.

7) “ Similar to the Gaussianity property ...” ? The importance of finding sources that are maximally non-Gaussian is not clear. The connection between sparseness, skewness, kurtosis and non-Gaussianity is not clear. Hyvarinen & Oja, 2000 does not mention sparseness? Why is a more sparse source considered to be a more correct source?

REPLY:

Thank you for your comment. To improve clarity, we have revised the statement as follows:

"Both algorithms rely on ICA to separate sources by maximizing a statistical measure of non-Gaussianity or sparsity of the estimated sources (Hyvarinen & Oja, 2000). In BSS of EMG signals, the main characteristics used for decomposition is sparseness (Farina & Holobar, 2016) since motor unit discharge times are not fully independent."

For a more in-depth discussion on the role of sparseness and its connection to source separation in EMG decomposition, we refer readers to Farina (2016), which provides a comprehensive review of the topic. While we acknowledge the importance of these concepts, we aim to keep the discussion focused, as they have been extensively covered in prior literature.

8) What is y ? Is it a scalar, representing one channel, or a vector, representing all channels?

REPLY:

Thank you for your comment. The variable should have been s , not y , representing the source vector.

9) How is the contrast function for each source determined and used?

REPLY:

Thank you for your comment. The contrast function is defined as $G(s) = E[\text{sign}(s)|s|^e]$ as described in the *Methods* section. This function introduces a non-linearity applied to the source vector, which provides a measure of sparsity and serves as our loss function.

10) It is not clear how the use of a different “optimal” contrast function for each source is beneficial.

REPLY:

Thank you for your comment. To clarify the benefit of using a different *optimal* contrast function for each source, we have added the following explanation:

"The core hypothesis is that each source requires a different level of selectivity to be effectively separated from the mixture. Sources that are more similar to one another may require stronger discriminants, whereas others may require less stringent criteria."

This addition explicitly highlights why adapting the contrast function to each source can improve decomposition performance. The reason why it's useful is reported in the *Results*.

11) " ... the loss $G(s)$.." should be " ... the last $G(s)$..."?

REPLY:

Thank you for your comment. The correct term is *loss $G(s)$* as $G(s)$ measures the sparsity of the source, which we aim to maximize. However, since the term *loss* was only used once in the manuscript, we have removed it to refer exclusively to the *contrast function* for consistency and clarity.

12) How is failure of a $G(s)$ to provide improvement determined?

REPLY:

Failure of $G(s)$ to provide improvement is determined by monitoring whether $G(s)$ ceases to increase. This stopping criterion ensures that the optimization process does not continue unnecessarily once no further improvement is observed.

To clarify this, we have explicitly stated this condition in the *Methods* section.

13) "After each ICA step ..." What was the output of each ICA step?

REPLY:

The output of each ICA step consists of the updated separation vectors and the optimal exponent, as well as an estimate of the source. To clarify this, we have revised the text as follows:

"After each ICA step, a peak-finding algorithm detected the source samples, as calculated with the updated separation vectors, ..."

This revision explicitly states the role of the separation vectors in determining the source samples.

14) " ... two-class k-medoid clustering to identify potential motor unit spikes." What was the input to the clustering algorithm? How are the discovered clusters used?

REPLY:

Thank you for your comment. To clarify, the input to the clustering algorithm consists of the estimated source, from which we want to optimally detect the peaks positions in time. We have revised the text as follows:

"This was followed by a two-class k-medoid clustering applied to the estimated sources to identify their peaks ..."

The discovered clusters are then used to differentiate motor unit spikes from noise, ensuring that only relevant spikes are retained for further optimisation.

15) What is are “potential motor unit spikes”?

REPLY:

Potential motor unit spikes refer to the detected motor unit firings at each iteration. They are termed *potential* because the decomposition process is iterative, and the identification of spikes is refined throughout successive ICA runs until the final decomposition is achieved.

16) “ ... using a fitness function based on the coefficient of variation of the interspike intervals ...” When studying unknown MN firing patterns, why is it useful to assume that their patterns will be maximally regular?

REPLY:

Thank you for your comment. The assumption of low variation in interspike intervals is useful as it serves as a *source quality metric*, helping to distinguish real motor unit activity from noise or spurious detections. The goal is not to assume *maximal* regularity but rather to ensure that the coefficient of variation remains within a reasonable range (typically <30%, (Holobar, A. et al., 2010)).

This differs from cortical neuron firing patterns, which often follow a Poisson distribution, as motor neurons in the peripheral nervous system exhibit more regular firing behaviour.

17) “ ... the regularity of motor unit firing in isometric contractions.” What about non-isotonic force contractions?

REPLY:

In non-isometric force contractions, regularity in motor unit firing cannot be assumed. This is why, for ballistic contractions, we used the silhouette value (*SIL*) instead of the coefficient of variation (*CoV*) as a source quality metric.

We had previously accidentally omitted this detail in the *Methods* section, but we have now added the following statement:

"When the regularity of firings could not be assumed, as in ballistic contractions, the SIL was used instead to assess source quality."

18) “After each ICA update ...” How was the ICA updated? What about the ICA was updated?

REPLY:

At each step, an ICA run was conducted independently on each separation vector. This iterative process ensured that the separation vectors were continuously refined, allowing for improved source extraction throughout the decomposition. This has been clarified in the *Methods*.

19) 20 iterations or ten updates which is it?

REPLY:

Thank you for your comment. The two numbers refer to different update processes:

- **20 iterations** refer to the stopping criterion for each **ICA update**—if the contrast function $G_SCD(s)$ stops increasing after 20 iterations, the ICA run is terminated early.
- **10 updates** refer to the **swarm updates**, which dictate how often the exponent adaptation occurs within the swarm optimization framework.

20) Was each source iteratively estimated one at a time, via the use of its optimal $G(s)$, followed by its “peeling-off”?

REPLY:

Yes, exactly.

21) What happens if a MUPT does not pass the silhouette test or meet the COV and firing rate criteria? Is it still peeled off?

REPLY:

Thank you for your comment. No, only sources that pass the criteria are peeled off. To clarify the peel off procedure, we have revised the text as follows:

"If the source passed this evaluation ($silhouette > 0.85$) and had not been identified in previous iterations, the motor unit action potentials were subtracted—i.e., peeled off—from the signal by creating a template motor unit waveform and removing it at each identified motor unit firing time. This subtraction prevented further convergence to the same source in subsequent decomposition iterations."

22) “ablation” does not seem to be the best term to use here?

REPLY:

Ablation refers to the removal of a specific component to assess its impact, which accurately describes our approach. In this case, we remove the swarm update and the peel-off mechanism to evaluate how their absence affects the algorithm’s performance.

Since this term is commonly used in machine learning and algorithmic studies for similar analyses, we believe it remains the most relevant choice in this context. However, we appreciate the reviewer’s consideration and are open to alternative suggestions if a different term is preferred.

23) “The innervated muscle fibers were randomly and uniformly distributed throughout the muscle volume ...” Were MU territories used?

REPLY:

Yes, MU territories were used. The size of the territory was determined based on the number of fibres in the motor unit. For further details, we refer the reader to *Neuromotion*, where these modeling principles are described in more depth (Ma et al., 2024a).

24) “ ... decreases in female individuals” Is this not a bigger problem when studying the quadriceps then the TA? If so, why were quadriceps studies not included?

REPLY:

Thank you for your comment. As previously mentioned, the primary focus of this study was the validation of the algorithm rather than an exhaustive investigation of different muscle groups.

While we acknowledge that the quadriceps pose additional challenges due to their depth and the possible large subcutaneous layers, testing the algorithm on all possible muscle groups was beyond the scope of this study. However, our decision to share the code openly allows researchers to apply the method to other muscle groups, including the quadriceps, in future studies.

25) “ ...such as when recording from deep muscles or from muscles covered by a large layer of subcutaneous fat ...” Again, why were quadriceps studies not included? The TA is quite a superficial muscle by comparison.

REPLY:

As previously mentioned, the primary focus of this study was the validation of the algorithm rather than an exhaustive investigation of different muscle groups.

While we acknowledge that the quadriceps pose additional challenges due to their depth and the presence of subcutaneous fat, testing the algorithm on all possible muscle groups was beyond the scope of this study. However, our decision to share the code openly allows researchers to apply the method to other muscle groups, including the quadriceps, in future studies.

Dear Professor Farina,

Re: JP-TFP-2025-287913R1 "Unlocking the Full Potential of High-Density Surface EMG: Novel Non-Invasive High-Yield Motor Unit Decomposition" by Agnese Grison, Irene Mendez Guerra, Alexander Kenneth Clarke, Silvia Muceli, Jaime Ibanez Pereda, and Dario Farina

We are pleased to tell you that your paper has been accepted for publication in The Journal of Physiology.

- You must start the Methods section with a paragraph headed Ethical Approval.

PLEASE NOTE = If experiments were conducted on humans, confirmation that informed consent was obtained, preferably in writing, that the studies conformed to the standards set by the latest revision of the Declaration of Helsinki and that the procedures were approved by a properly constituted ethics committee, which should be named, must be included in the article file. If the research study was registered (clause 35 of the Declaration of Helsinki), the registration database should be indicated, otherwise the lack of registration should be noted as an exception (e.g. The study conformed to the standards set by the Declaration of Helsinki, except for registration in a database). For further information see: <https://physoc.onlinelibrary.wiley.com/hub/human-experiments>.

Authors should note that it is too late at this point to offer corrections prior to proofing. Major corrections at proof stage, such as changes to figures, will be referred to the Editors for approval before they can be incorporated. Only minor changes, such as to style and consistency, should be made at proof stage. Changes that need to be made after proof stage will usually require a formal correction notice.

All queries at proof stage should be sent to: TJP@wiley.com

If you would like to receive our 'Research Roundup', a monthly newsletter highlighting the cutting-edge research published in The Physiological Society's family of journals (The Journal of Physiology, Experimental Physiology and Physiological Reports), please click this link, fill in your name and email address and select 'Research Roundup': <https://www.physoc.org/journals-and-media/membernews/>

Yours sincerely,

Richard Carson
Senior Editor
The Journal of Physiology

P.S. - You can help your research get the attention it deserves! Check out Wiley's free Promotion Guide for best-practice recommendations for promoting your work at www.wileyauthors.com/eoo/guide. You can learn more about Wiley Editing Services which offers professional video, design, and writing services to create shareable video abstracts, infographics, conference posters, lay summaries, and research news stories for your research at www.wileyauthors.com/eoo/promotion.

IMPORTANT NOTICE ABOUT OPEN ACCESS: To assist authors whose funding agencies mandate public access to published research findings sooner than 12 months after publication The Journal of Physiology allows authors to pay an Open Access (OA) fee to have their papers made freely available immediately on publication.

You can check if your funder or institution has a Wiley Open Access Account here: <https://authorservices.wiley.com/author->

EDITOR COMMENTS

Reviewing Editor:

All reviewers' comments have been satisfactorily addressed.

Senior Editor:

Comments to the Author:

Please consider the recommendations provided by Referee #1 when preparing the final version of the manuscript.

REFEREE COMMENTS:

Referee #1:

General: I appreciate the authors' detailed reply to every comment from the original review. Based on the changes made, I have no further general comments/concerns. I include a few minor grammatical comments.

Specific:

- 1) Text of graphical abstract: Suggest "... output are..." should either be "... output is ..." or "outputs are ...".
- 2) Introduction, first sentence. You have both the British spelling of "fibre" and the U.S. spelling of "fiber" in the same sentence. Most journals request that you pick one style and use it throughout.
- 3) P 10, bottom. Since CMRR varies with frequency, it is generally specified at the power-line frequency, i.e., "... common mode rejection ratio >95 dB at 50 Hz...".
- 4) In the MARK_UP version of the manuscript, it appears that sections 3.4, 3.5 and 3.6 each list "Pilot 1" (instead of "Pilot 1", "Pilot 2", and "Pilot 3'). But, the clean-copy version does not seem to have this apparent error. Please confirm that the clean-copy sections are correctly titled.

Referee #2:

The revised manuscript suitably addresses my previous comments.